# The cardiopharyngeal mesoderm contributes to lymphatic vessel development in mouse

Kazuaki Maruyama[1,2]*, Sachiko Miyagawa-Tomita[1,3,4], Yuka Haneda[1], Mayuko Kida[1], Fumio Matsuzaki[5], Kyoko Imanaka-Yoshida[2], Hiroki Kurihara[1]*

[1]Department of Physiological Chemistry and Metabolism, Graduate School of Medicine, The University of Tokyo, Bunkyo-ku, Japan; [2]Department of Pathology and Matrix Biology, Graduate School of Medicine, Mie University, Mie, Japan; [3]Department of Animal Nursing Science, Yamazaki University of Animal Health Technology, Tokyo, Japan; [4]Heart Center, Department of Pediatric Cardiology, Tokyo Women's Medical University, Tokyo, Japan; [5]Laboratory for Cell Asymmetry, RIKEN Center for Biosystems Dynamics Research, Kobe, Japan

*For correspondence:
k.maruyama0608@gmail.com
(KM);
kuri-tky@umin.net (HK)

Competing interest: The authors declare that no competing interests exist.

**Abstract** Lymphatic vessels are crucial for tissue homeostasis and immune responses in vertebrates. Recent studies have demonstrated that lymphatic endothelial cells (LECs) arise from both venous sprouting (lymphangiogenesis) and de novo production from non-venous origins (lymphvasculogenesis), which is similar to blood vessel formation through angiogenesis and vasculogenesis. However, the contribution of LECs from non-venous origins to lymphatic networks is considered to be relatively small. Here, we identify the *Islet1* (*Isl1*)-expressing cardiopharyngeal mesoderm (CPM) as a non-venous origin of craniofacial and cardiac LECs. Genetic lineage tracing with *Isl1*^Cre/+ and *Isl1*^CreERT2/+ mice suggested that a subset of CPM cells gives rise to LECs. These CPM-derived LECs are distinct from venous-derived LECs in terms of their developmental processes and anatomical locations. Later, they form the craniofacial and cardiac lymphatic vascular networks in collaboration with venous-derived LECs. Collectively, our results demonstrate that there are two major sources of LECs, the cardinal vein and the CPM. As the CPM is evolutionarily conserved, these findings may improve our understanding of the evolution of lymphatic vessel development across species. Most importantly, our findings may provide clues to the pathogenesis of lymphatic malformations, which most often develop in the craniofacial and mediastinal regions.

## Editor's evaluation

This paper provides fundamental insight into the developmental source of lymphatic endothelial cells, which has been debated for over a century. This important work characterises the development of the mouse craniofacial lymphatics, and provides compelling evidence for a non-venous source of craniofacial lymphatic endothelial cells. The manuscript is well presented and will be of interest to developmental, vascular and lymphatic vascular biologists.

## Introduction

The lymphatic vascular system plays diverse roles in the maintenance of tissue fluid balance, immune surveillance, lipid absorption from the gut, and tumor metastasis (*Oliver and Alitalo, 2005*). Furthermore, recent progress in molecular and cellular characterization has unveiled the processes involved in the development of the lymphatic vasculature and the roles played by the lymphatic vasculature in

various pathophysiological conditions (*Klaourakis et al., 2021*; *Maruyama et al., 2021*; *Maruyama and Imanaka-Yoshida, 2022*; *Oliver et al., 2020*; *Oliver and Alitalo, 2005*).

The origin of lymphatic endothelial cells (LECs) has been discussed since the 1900s (*Huntington and McClure, 1910*; *Sabin, 1902*). Recent studies using genetic lineage tracing confirmed Sabin's hypothesis that lymphatic vessels originate from the embryonic cardinal vein through lymphangiogenesis (*Srinivasan et al., 2007*; *Yang et al., 2012*). In mice, lymphatic vessels arise from the common cardinal vein between embryonic day (E) 9.5 and E10.0 during the partial expression of prospero homeobox transcription protein 1 (Prox1), which is the master regulator of LEC differentiation (*Srinivasan et al., 2007*; *Wigle and Oliver, 1999*). After that, vascular endothelial growth factor C induces the sprouting and expansion of vascular endothelial growth factor receptor 3 (VEGFR3)$^+$ LECs from the common cardinal veins to form the first lymphatic plexus (*Hägerling et al., 2013*; *Karkkainen et al., 2004*). In contrast, in tadpoles and avian embryos, mesenchymal cells also contribute to lymphatic vessel development (*Ny et al., 2005*; *Schneider et al., 1999*; *Wilting et al., 2006*), supporting Huntington and McClure's suggestion that lymphatic vessels form via the differentiation of mesenchymal cells into LECs, which later develop into a primary lymph sac and connect to the venous system. Supporting these findings, recent studies involving various genetic lineage-tracing models have revealed that non-venous sources of LECs also contribute to the lymphatic vasculature in the skin, mesentery, and heart through lymphvasculogenesis (*Klotz et al., 2015*; *Lioux et al., 2020*; *Mahadevan et al., 2014*; *Martinez-Corral et al., 2015*; *Maruyama et al., 2019*; *Pichol-Thievend et al., 2018*; *Stanczuk et al., 2015*). Despite the accumulation of studies on non-venous sources of LECs, it remains uncertain how much they contribute to lymphatic vessels throughout the body. Our previous studies showed that LIM-homeodomain protein Islet1 (*Isl1*)-expressing second heart field cells contribute to ventral cardiac lymphatic vessels as a non-venous source of LECs (*Maruyama et al., 2019*). *Isl1*-expressing second heart field cells have been found to overlap with the progenitor populations that give rise to the pharyngeal muscles in mice and chicks, and they are collectively known as the cardiopharyngeal mesoderm (CPM), which contributes to broad regions of the heart, cranial musculature, and connective tissue (*Diogo et al., 2015*; *Grimaldi et al., 2022*; *Harel et al., 2009*; *Heude et al., 2018*; *Tirosh-Finkel et al., 2006*; *Tzahor and Evans, 2011*). The CPM is composed of the paraxial and splanchnic mesodermal cells surrounding the pharynx. Later, the CPM cells migrate to form the mesodermal cores of the pharyngeal arches. Before their differentiation, CPM cells express both immature skeletal and cardiac muscle markers, including *Isl1*, myocyte-specific enhancer factor 2C (*Mef2c*), and T-box transcription factor (*Tbx1*). In mice, these molecules have been shown to be markers of a subset of CPM cells, which play important roles in cardiovascular and skeletal muscle development (*Adachi et al., 2020*; *Diogo et al., 2015*; *Grimaldi et al., 2022*; *Harel et al., 2009*; *Heude et al., 2018*; *Lescroart et al., 2010*; *Nathan et al., 2008*; *Tzahor and Evans, 2011*).

Herein, we identified that the CPM contributes to broader regions of the facial, laryngeal, and cardiac lymphatic vasculature using *Isl1$^{Cre/+}$* and *Isl1$^{CreERT2/+}$* mice, which can be used to study developmental processes involving the CPM. According to our developmental analysis, CPM-derived progenitor cells have the capacity to differentiate into LECs for a limited embryonic period. CPM-derived LECs are spatiotemporally distinct from venous-derived LECs during the early embryonic period. Later, they coordinate to form capillary lymphatics in and around the cranial and cardiac regions. Conditional knockout (KO) of *Prox1* in *Isl1*-expressing cells resulted in decreased numbers of lymphatic vessels in the tongue and altered the proportions of *Isl1$^+$* and *Isl1$^-$* LECs in facial lymphatic vessels. These results suggest that a subpopulation of LECs may share a common mesodermal origin with cardiopharyngeal components. In addition, as the CPM is conserved across vertebrates, they may provide clues regarding the evolution of lymphatic vessel development. From a clinical viewpoint, head and neck regions contributed by the CPM are the most common sites of lymphatic malformations (LMs) (*Perkins et al., 2010*). Collectively, the present findings provide a fundamental basis for our understanding of lymphatic vessel development and lymphatic system-related diseases.

# Results

## *Isl1*+ CPM-derived LECs contribute to cardiac, facial, and laryngeal lymphatic vessel development

To assess the regional contribution of *Isl1*+ CPM cells to lymphatic vessel development, we crossed *Isl1*$^{Cre/+}$ mice, which express Cre recombinase under the control of the *Isl1* promoter and in which second heart field derivatives are effectively labeled, with the transgenic reporter line *Rosa26*$^{tdTomato/+}$ and analyzed at E16.5, when lymphatic networks are distributed throughout the whole body (**Srinivasan et al., 2007**). Co-immunostaining of platelet endothelial cell adhesion molecule (PECAM) and VEGFR3, which we confirmed its colocalization with lymphatic vessel endothelial hyaluronan receptor 1 (LYVE1) at E14.5 and E16.5 (**Figure 1—figure supplement 1**), revealed tdTomato+ LECs in and around the larynx, the skin of the lower jaw, the tongue, and the cardiac outflow tracts, at various frequencies, whereas no such cells were found on the dorsal side of the ventricles, which agrees with our previous study (**Maruyama et al., 2019**; **Figure 1A–H**). We also confirmed that tdTomato+ cells were present in populations of PECAM+ blood and aortic endothelial cells in these regions (**Figure 1A–F**).

To examine whether the *Isl*-marked LECs originated from the neural crest (**Engleka et al., 2012**), we used *Wnt1-Cre* mice, in which neural crest cells were effectively marked, crossed with the *Rosa26*$^{eYFP/+}$ reporter line. We detected broad eYFP+ cell contributions to the bone, cartilage, and mesenchymal tissues in the head and neck regions as well as the cardiac outflow tract wall, but no eYFP+ LECs were found in these regions (**Figure 1I–O**), excluding the possibility of them originating from the neural crest. We then investigated the possible contribution of myogenic CPM populations (**Harel et al., 2012**; **Harel et al., 2009**; **Heude et al., 2018**) to LECs using *Myf5*$^{CreERT2/+}$ mice crossed with the *Rosa26*$^{tdTomato/+}$ line. After tamoxifen was administered at E8.5, tdTomato+ cells were broadly detected in the skeletal muscle in the head and neck regions at E16.5, indicating effective Cre recombination in CPM-derived musculatures (**Figure 1—figure supplement 2A–E**). By contrast, no tdTomato+ LECs were detected in the cardiopharyngeal region (**Figure 1—figure supplement 2A–E**). These results indicate that *Isl1*+ CPM cells contribute to LECs in the cardiopharyngeal region through progenitors distinct from the *Myf5*+ myogenic lineage.

## *Isl1*+ CPM cells can differentiate into LECs in a limited developmental period

To determine the timing of the differentiation of CPM into LECs, we employed tamoxifen-inducible *Isl1*$^{CreERT2/+}$ mice crossed with either the *Rosa26*$^{tdTomato/+}$ or *Rosa26*$^{eYFP/+}$ reporter line to conditionally label descendants of *Isl1*-expressing cells. We intraperitoneally injected tamoxifen into pregnant female mice at several timepoints. We first injected tamoxifen at E6.5 and analyzed the embryos at E16.5 by examining sagittal sections that had been immunostained for PECAM and VEGFR3. We injected tamoxifen into three pregnant female mice; however, due to the toxicity of tamoxifen, we could only obtain two embryos from one mouse. In the embryos, tdTomato+ cells broadly contributed to LECs in and around the larynx, the skin of the lower jaw, the tongue, and the cardiac outflow tracts at various frequencies, but not in the back skin or on the dorsal side of the ventricles (**Figure 2—figure supplement 1A–I**). We then injected tamoxifen at E8.5 and analyzed the embryos at E16.5. Although the contribution of tdTomato+ cells to LECs was smaller, tdTomato+ cells broadly contributed to LECs, as was found for the tamoxifen-treated embryos analyzed at E6.5 (**Figure 2A–H**). In contrast, the number of tdTomato+ LECs seen at E16.5 was markedly decreased when tamoxifen was injected at E11.5 (**Figure 2I–P**). These results indicate that the differentiation of CPM cells into LECs occurs before E11.5.

We then performed whole-mount immunostaining of hearts from E16.5 *Isl1*$^{CreERT2/+}$;*Rosa26*$^{eYFP/+}$ embryos for PECAM, VEGFR3, and eYFP to show the spatial contribution of *Isl1*+ CPM cells to lymphatic vessels. In the embryos from the mice treated with tamoxifen at E8.5, eYFP+ cells were detected in VEGFR3+ lymphatic vessels around the cardiac outflow tracts. In contrast, we could not detect eYFP+ cells in the VEGFR3+ lymphatic vessels around the cardiac outflow tracts when tamoxifen was administered at E11.5, or on the dorsal side of the ventricles when tamoxifen was administered at E8.5 or E11.5 (**Figure 3A–H**). These results indicate that *Isl1*+ CPM cells gradually lose their capacity to differentiate into LECs as development progresses.

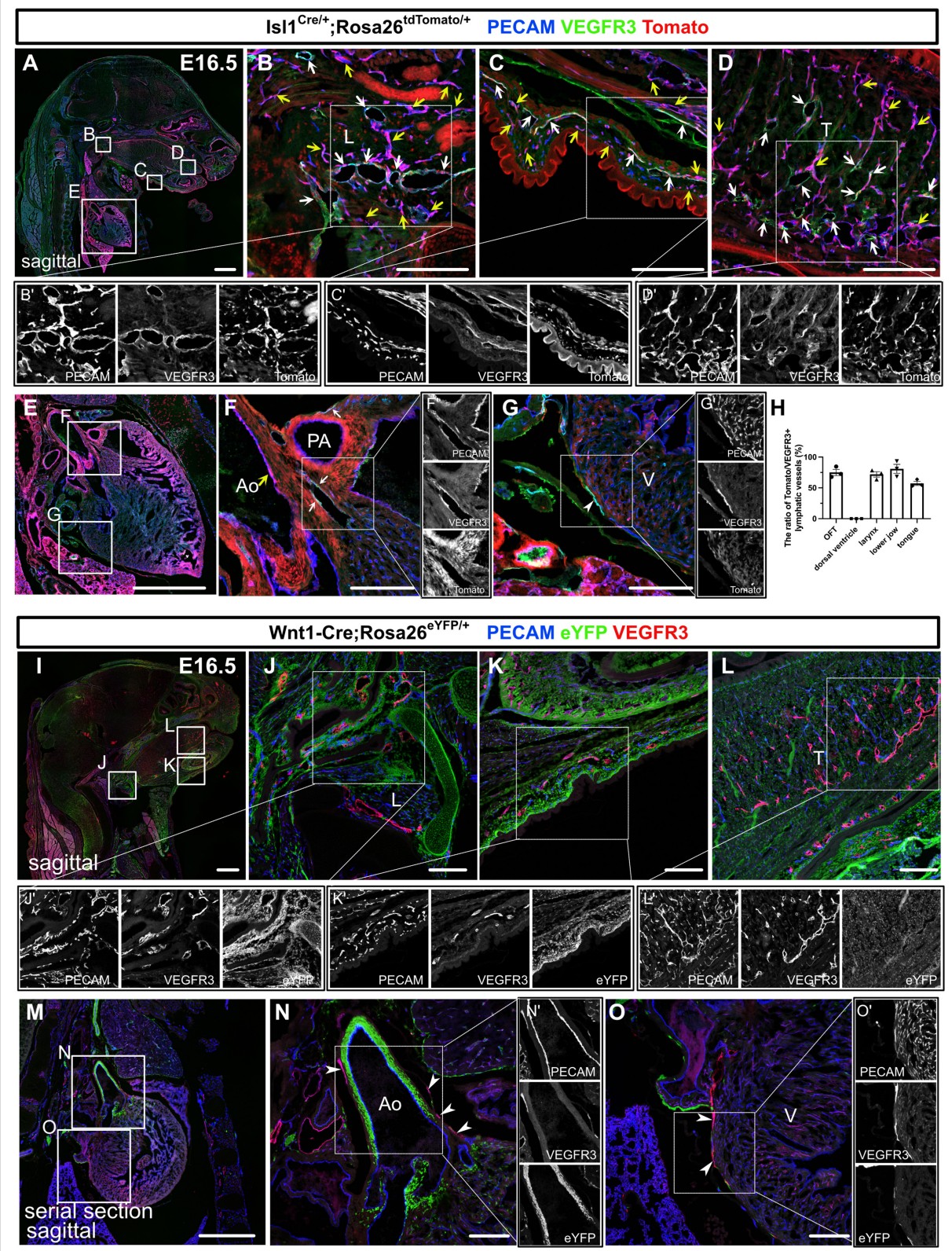

**Figure 1.** *Isl1*[+] lineages contribute to cranial and cardiac lymphatic vessels. (**A–G′**) Sagittal sections of *Isl1*[Cre/+]*;Rosa26*[tdTomato/+] embryos, in which platelet endothelial cell adhesion molecule (PECAM), tdTomato, and vascular endothelial growth factor receptor 3 (VEGFR3) were labeled at embryonic day (E) 16.5, are shown. (**A–F**) tdTomato colocalized with PECAM/VEGFR3 (white arrows) and PECAM (yellow arrows) in and around the larynx (**B**), the skin of the lower jaw (**C**), the tongue (**D**), and the cardiac outflow tracts (**F**) (n=3). (**G**) tdTomato did not colocalize with VEGFR3 in the LECs on the dorsal side of the

*Figure 1 continued on next page*

*Figure 1 continued*

ventricles (white arrowhead, n=3). (**H**) The results of quantitative analysis of the percentage of the area in tdTomato⁺/VEGFR3⁺ lymphatic vessels among all VEGFR3⁺ lymphatic vessels are shown. (**I–O'**) Sagittal sections of *Wnt1-Cre;Rosa26^eYFP/+* embryos, in which PECAM, eYFP, and VEGFR3 were labeled at E16.5, are shown. There were no eYFP⁺/VEGFR3⁺ lymphatic vessels in or around the larynx, the skin of the lower jaw, the tongue, or the heart (the white arrowheads indicate cardiac lymphatic vessels around the cardiac outflow tracts and the dorsal side of the ventricles; n=3). Each dot represents a value obtained from one sample. L, larynx; T, tongue; Ao, aorta; PA, pulmonary artery; V, ventricle. Scale bars, 100 µm (**B–D, F, G, J–L, N, O**), 1 mm (**A, E, I, M**).

The online version of this article includes the following source data and figure supplement(s) for figure 1:

**Source data 1.** Quantification of Isl1⁺ lineages contribution to craniofacial and cardiac lymphatic vessels.

**Figure supplement 1.** Lymphatic vessel endothelial hyaluronan receptor 1 (LYVE1) and vascular endothelial growth factor receptor 3 (VEGFR3) are co-expressed in lymphatic vessels in the tongue, facial skin, and the outflow tracts.

**Figure supplement 2.** *Myf5⁺* lineages do not generate lymphatic endothelial cells (LECs) in the cranial or cardiac regions.

## *Isl1⁺* CPM-derived LECs are spatiotemporally distinct from vein-derived LECs

We next analyzed the early stages of the spatiotemporal development of *Isl1⁺* CPM-derived lymphatic vessels using *Isl1^Cre/+;Rosa26^eYFP/+* mice by performing immunostaining for PECAM, Prox1, and eYFP. At E11.5, eYFP⁺/Prox1⁺ cells were detected around the eYFP⁺ mesodermal core region of the first and second pharyngeal arches in *Isl1^Cre/+;Rosa26^eYFP/+* mice (**Figure 4A and B**). At E12.0, eYFP⁺/Prox1⁺ cells were distributed in the pharyngeal arch region and extended toward the lymph sac-forming domain composed of eYFP⁻/Prox1⁺ cells, which were considered to emerge from the cardinal vein and inter-somitic vessels, as reported previously (*Yang et al., 2012*; *Figure 4C and D*). At E14.5, eYFP⁺/Prox1⁺/PECAM⁺ LECs formed lymphatic capillaries in the lower jaw and the tongue, which are derived from the first pharyngeal arch (*Figure 4E–G*).

We then analyzed *Isl1^CreERT2/+;Rosa26^eYFP/+* embryos at E12.0. After tamoxifen was administered at E8.5, eYFP⁺/Prox1⁺ cells were found in and around the pharyngeal mesodermal condensation, close to the lymph sac-forming region, as was seen in the *Isl1^Cre/+;Rosa26^eYFP/+* embryos (*Figure 4H, I*). No eYFP⁺/Prox1⁺ cells were observed in or around the cardinal veins (*Figure 4J*). Some of the eYFP⁺/Prox1⁺ cells in the pharyngeal arch region expressed PECAM at E12.0, indicating that they had endothelial characteristics (*Figure 4K and L*). After tamoxifen was administered at E9.5, the number of eYFP⁺/Prox1⁺ LECs was significantly decreased compared with that seen in the embryos treated with tamoxifen at E8.5 (*Figure 4M–O*).

To identify possible *Isl1⁺* LEC progenitors, we investigated the expression patterns of Isl1, Prox1, and vascular endothelial markers (Flk1 and PECAM) by immunostaining sections of E9.0–E11.5 pharyngeal arches and cardinal veins. Consistent with the previous report (*Cai et al., 2003*), Isl1 was abundantly expressed in the core mesoderm of the first and second pharyngeal arches corresponding to the CPM from E9.0 to E11.5 (*Nathan et al., 2008*), where Prox1⁺ cells also aggregated and partially overlapped with Isl1 signals (*Figure 3—figure supplement 1A, A' C, C' E, E' G, G' I, I'*). By contrast, Flk1⁺ or PECAM⁺ cells were distributed mainly around the CPM and not expressed Isl1 (*Figure 3—figure supplement 1A, A' C, C' E, E' G, G' I, I'*). Furthermore, Isl1 was expressed neither in the endothelial layer of the cardinal vein nor in surrounding Prox1⁺/PECAM⁺ LECs (*Figure 3—figure supplement 1B, B' D, D', F, F', H, H', J, and J'*). Taken together with the result from *Myf5^CreERT2/+* mice, these results indicate that *Isl1⁺* non-myogenic CPM cells may serve as LEC progenitors independent of venous-derived LECs and the commitment to LEC differentiation occurs before E9.5 in the pharyngeal arch region.

## Loss of Prox1 in the *Isl1⁺* lineage reveals the importance of *Isl1⁺* CPM-derived LECs for cranial lymphatic vessel development

To investigate the functional importance of *Isl1⁺* CPM-derived LECs for the development of cranial lymphatic vessels (the lymphatics of the lower jaw skin, cheeks, and tongue), we conditionally knocked out *Prox1* in CPM populations by crossing *Prox1*-flox (*fl*) mice, which expressed eGFP under the control of the *Prox1* promoter upon Cre-mediated exon 2 deletion (*Iwano et al., 2012*), with *Isl1^Cre/+* mice. When Prox1 is knocked down in the *Tek⁺* lineage, an initial failure in specification of LECs was confirmed at E14.5 with a lack of LECs even at E17.5 (*Klotz et al., 2015*; *Lioux et al., 2020*; *Maruyama et al.,*

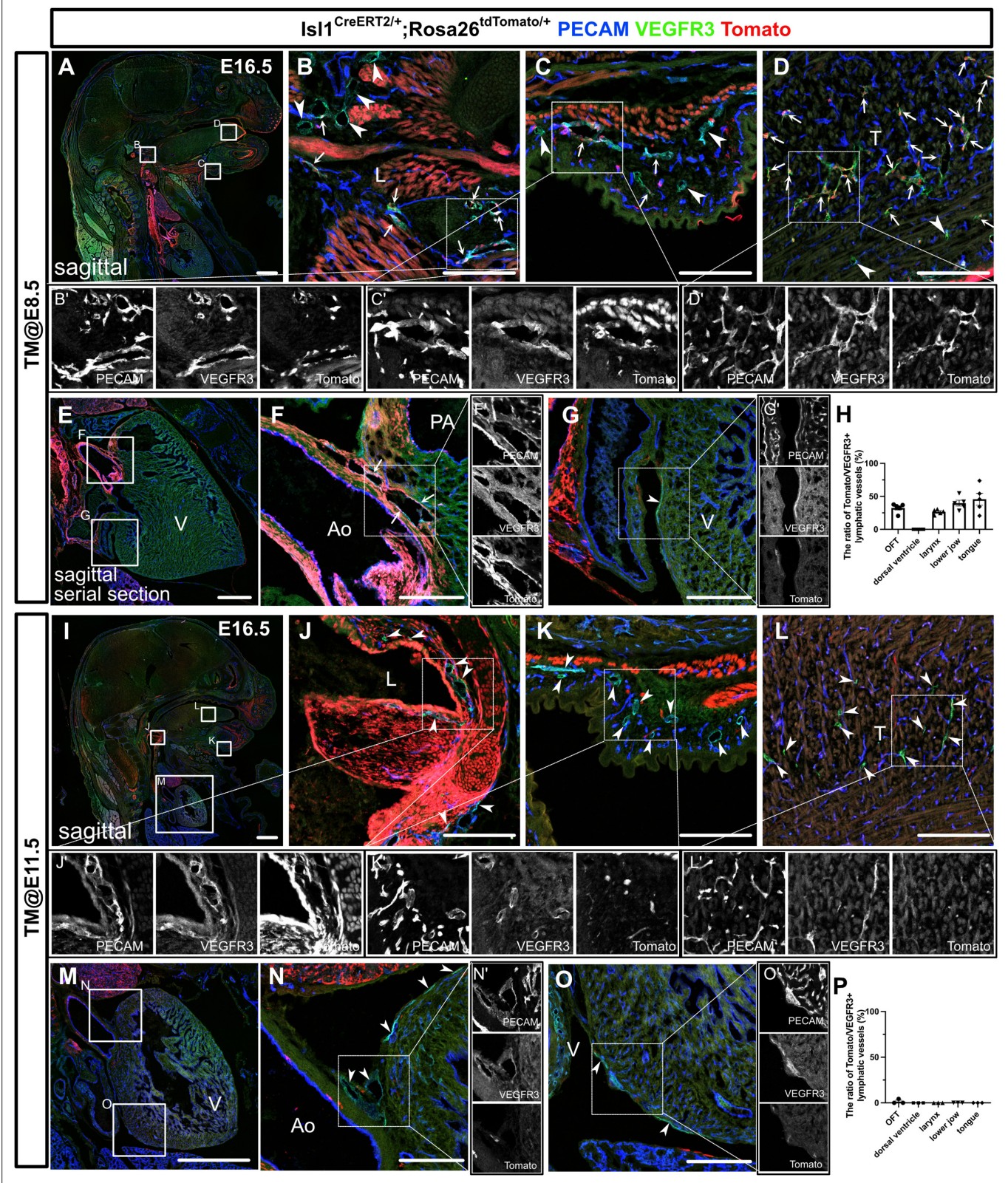

**Figure 2.** The differentiation of *Isl1*[+] cardiopharyngeal mesoderm (CPM) cells into lymphatic endothelial cells (LECs) occurs during the early embryonic period. (**A–G′, I–O′**) Sagittal sections of *Isl1^CreERT2/+^;Rosa26^tdTomato/+^* embryos, in which platelet endothelial cell adhesion molecule (PECAM), tdTomato, and vascular endothelial growth factor receptor 3 (VEGFR3) were labeled at embryonic day (E) 16.5, are shown. Tamoxifen was administered at E8.5 (**A–G′**) or E11.5 (**I–O′**). (**A–G′**) Both tdTomato[+] (white arrows) and tdTomato[-] (white arrowheads) VEGFR3[+] lymphatic vessels were observed in and around

*Figure 2 continued on next page*

*Figure 2 continued*

the larynx (**B**), the skin of the lower jaw (**C**), the tongue (**D**), and the cardiac outflow tracts (**F**) (n=5). (**G**) tdTomato did not colocalize with VEGFR3 in the LECs on the dorsal side of the ventricles (white arrowhead, n=5). (**I–O'**) Almost all of the VEGFR3[+] lymphatic vessels in and around the larynx (**J**), the skin of the lower jaw (**K**), the tongue (**L**), the cardiac outflow tracts (**N**), and the dorsal side of the ventricles (**O**) were tdTomato[-] (white arrows, n=3). (**H, P**) The results of a quantitative analysis of the percentage of the area of tdTomato[+]/VEGFR3[+] lymphatic vessels among all VEGFR3[+] lymphatic vessels are shown. Tamoxifen was administered at E8.5 (**H**) or E11.5 (**P**). Each dot represents a value obtained from one sample. L, larynx; T, tongue; Ao, aorta; PA, pulmonary artery; V, ventricle. Scale bars, 100 μm (**B–D, F, G, J–L, N, O**), 1 mm (**A, E, I, M**).

The online version of this article includes the following source data and figure supplement(s) for figure 2:

**Source data 1.** Quantification of Isl1[+] lineages contribution to craniofacial and cardiac lymphatic vessels at embryonic day (E) 16.5 with tamoxifen treatment at E8.5.

**Source data 2.** Quantification of Isl1[+] lineages contribution to craniofacial and cardiac lymphatic vessels at embryonic day (E) 16.5 with tamoxifen treatment at E11.5.

**Figure supplement 1.** The *Isl1[+]* cardiopharyngeal mesoderm (CPM) broadly contributes to cranial and cardiac lymphatic vessels.

**Figure supplement 1—source data 1.** Quantification of *Isl1[+]* lineages contribution to craniofacial and cardiac lymphatic vessels at embryonic day (E) 16.5 with tamoxifen treatment at E6.5.

*2019*). Therefore, we compared lymphatic vessel phenotypes at E16.5, by which systemic lymphatics formation is normally completed (*Srinivasan et al., 2007*). To test the recombination efficiency of the Prox1 locus in *Isl1[+]* LECs, we performed whole-mounted and section immunostaining with PECAM, eGFP, and Prox1 in *Isl1^{Cre/+};Prox1^{fl/+}* heterozygous and *Isl1^{Cre/+};Prox1^{fl/fl}* homozygous mice at E12.5 and

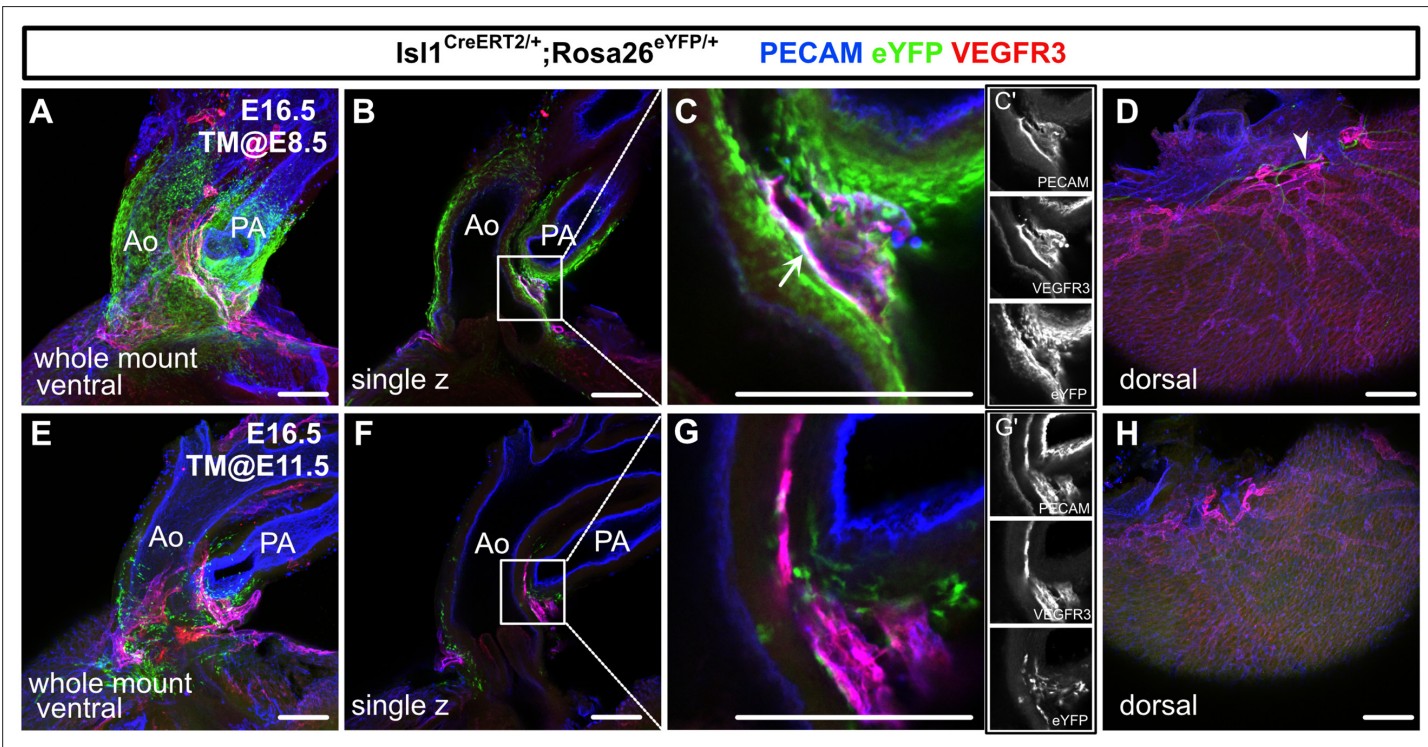

**Figure 3.** *Isl1[+]* cardiopharyngeal mesoderm (CPM) cells contribute to cardiac lymphatic vessel development. (**A–H**) Whole-mount confocal images of *Isl1^{CreERT2/+};Rosa26^{eYFP/+}* hearts, in which platelet endothelial cell adhesion molecule (PECAM), eYFP, and vascular endothelial growth factor receptor 3 (VEGFR3) were labeled at embryonic day (E) 16.5, are shown. Tamoxifen was administered at E8.5 (**A–D**) or E11.5 (**E–H**). (**A–D**) Many eYFP[+] cells were observed around the cardiac outflow tracts, and they contributed to lymphatic vessels (white arrow) (n=9/9 hearts [100%]) (**A–C'**). (**D**) There were no eYFP[+] lymphatic vessels on the dorsal side of the ventricles (n=9/9 hearts [100%]). The cardiac nerves were also positive for eYFP after tamoxifen treatment at E8.5 (white arrowhead) (**D**). (**E–H**) Fewer eYFP[+] cells were observed around the cardiac outflow tracts (**E–G'**), and there were no eYFP[+] lymphatic vessels around the cardiac outflow tracts (**F**) or on the dorsal side of the ventricles (n=0/8) (**H**). Ao, aorta; PA, pulmonary artery. Scale bars, 100 μm (**A–C, E–G**), 500 μm (**D, H**).

The online version of this article includes the following figure supplement(s) for figure 3:

**Figure supplement 1.** Isl1 is not expressed in the endothelium in the first and second pharyngeal arches and cardinal vein at early embryonic period.

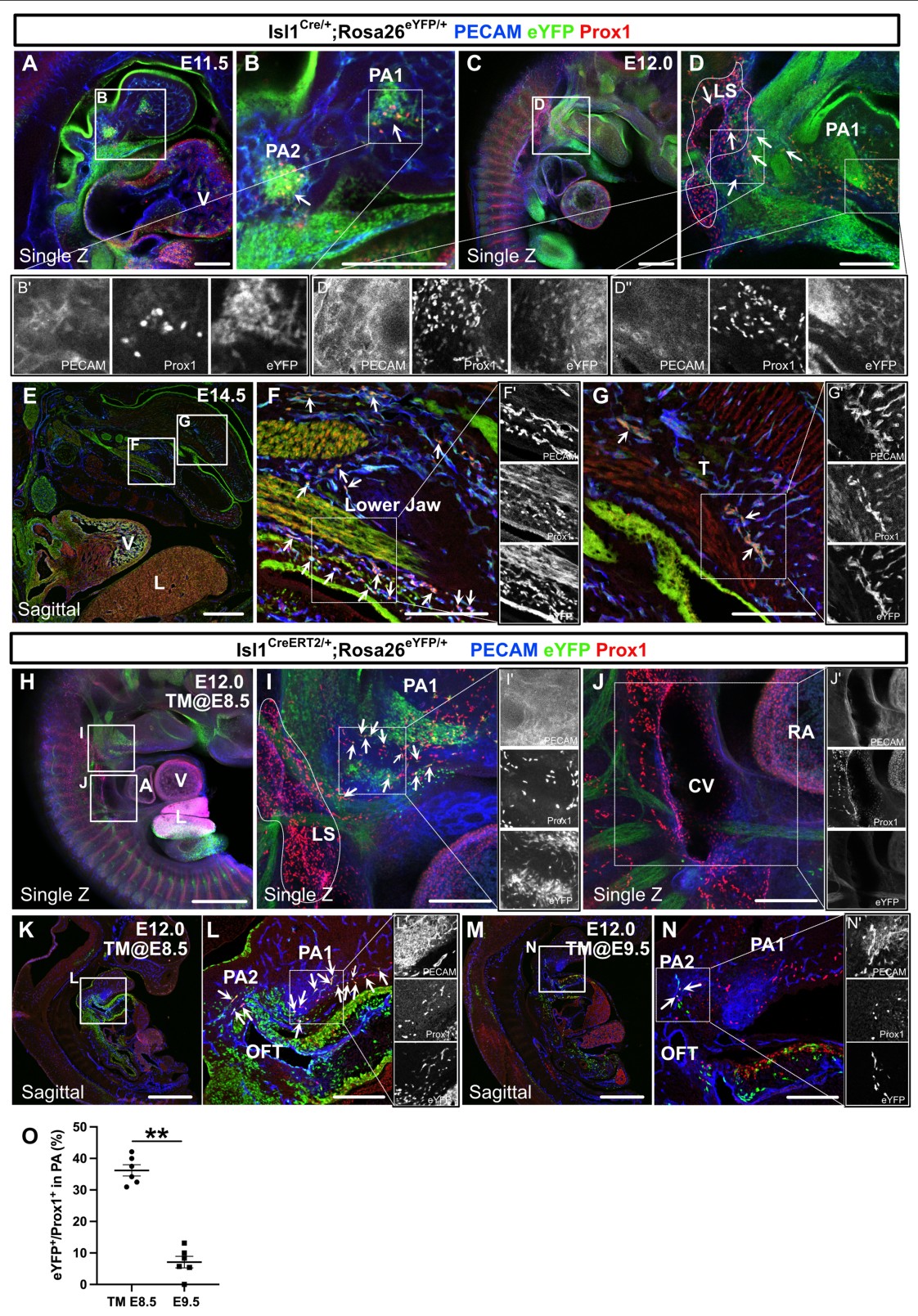

**Figure 4.** The spatiotemporal development of *Isl1*[+] cardiopharyngeal mesoderm (CPM)-derived lymphatic endothelial cells (LECs). (**A–G′**) Whole-mount and sagittal section images of *Isl1*[Cre/+];*Rosa26*[eYFP/+] embryos, in which platelet endothelial cell adhesion molecule (PECAM), eYFP, and Prox1 were labeled at embryonic day (E) 11.5, E12.0, or E14.5, are shown. (**A, B**) eYFP[+]/Prox1[+] cells were observed around the cores of the first and second pharyngeal arches at E11.5 (white arrowheads). (**C–D″**) eYFP[+]/Prox1[+] LECs migrated from the first and second pharyngeal arches to the lymph sac-

*Figure 4 continued on next page*

*Figure 4 continued*

forming domain (white dotted region) adjacent to the anterior cardinal vein at E12.0 (white arrows). (**E–G'**) Some of the eYFP⁺/Prox1⁺ LECs expressed PECAM and formed lymphatic capillaries in the lower jaw and tongue at E14.5 (white arrows). (**H–N'**) Whole-mount and sagittal section images of *Isl1^CreERT2/+*;*Rosa26^eYFP/+* embryos, in which PECAM, eYFP, and Prox1 were labeled at E12.0, are shown. Tamoxifen was administered at E8.5 (**H–L'**) or E9.5 (**M–N'**). (**I, I'**) eYFP⁺/Prox1⁺ LECs (white arrows) migrated from the first and second pharyngeal arches to the lymph sac-forming domain (white dotted region) at E12.0. (**J**) There were no eYFP⁺/Prox1⁺ cells in or around the cardinal vein (n=4). (**K–N**) eYFP⁺/Prox1⁺/PECAM⁺ LECs were seen in the first and second pharyngeal arches of the embryos at E12.0, when tamoxifen was administered at E8.5 or E9.5 (white arrows), although the number of these cells was decreased in the E9.5 group (**M–N'**). (**O**) The results of a quantitative analysis of the percentage of eYFP⁺/Prox1⁺ cells among Prox1⁺ cells in the first and second pharyngeal arches at E12.0 after tamoxifen treatment at E8.5 (the number of eYFP⁺/Prox1⁺ cells [10.83 (mean)±1.249 (SEM)]: Prox1⁺ cells [30.83±4.549]) or E9.5 (the number of eYFP⁺/Prox1⁺ cells [2.833±1.108]: Prox1⁺ cells [35.50±5.847]) are shown. \*\*p=0.0022. All of the data are presented as the mean ± SEM, and statistical analyses were performed using the non-parametric Mann-Whitney U-test. V, ventricle; PA1, first pharyngeal arch; PA2, second pharyngeal arch; CV, cardinal vein; LS, lymph sac-forming domain; L, liver; T, tongue; RA, right atrium; OFT, cardiac outflow tract. Scale bars, 100 μm (**A, B, D, F, G, I, J, L, N**), 500 μm (**C, E, H, K, M**); \*\*p<0.01.

The online version of this article includes the following source data for figure 4:

**Source data 1.** Quantification of eYFP⁺/Prox1⁺ cells among Prox1⁺ cells in the first and second pharyngeal arches at embryonic day (E) 12.0 with tamoxifen treatment at E8.5 and E9.5.

E13.5. At E12.5, eGFP⁺/Prox1⁺ cells were observed in the PA1 of *Isl1^Cre/+*;*Prox1^fl/+* heterozygous mice, whereas the number of Prox1⁺ cells was decreased and most of eGFP⁺ cells were negative for Prox1 in *Isl1^Cre/+*;*Prox1^fl/fl* homozygous mice (**Figure 5—figure supplement 1A–F**), indicating efficient knockdown of Prox1. At E13.5, eGFP⁺/Prox1⁺ cells were almost diminished in the tongue, whereas eGFP⁺/Prox1⁺ cells were still observed in the lower jaw in *Isl1^Cre/+*;*Prox1^fl/fl* homozygous mice (**Figure 5—figure supplement 1G–P**). This discrepancy may indicate that the recombination efficiency differs among tissues and that embryos with low recombination efficiency could survive until E13.5. When lymphatic vessel formation in these regions was analyzed in mice that were heterozygous or homozygous for the *Prox1^fl* allele, the number and area of VEGFR3⁺/PECAM⁺ lymphatic vessels were significantly lower in the tongues of the *Isl1^Cre/+*;*Prox1^fl/fl* homozygous mice than in those of the *Isl1^Cre/+*;*Prox1^fl/+* heterozygous mice (**Figure 5A–C, F–H, K and L**). In contrast, the formation of facial lymphatic vessels in the lower jaw and cheeks was not significantly affected by the deletion of *Prox1* in the *Isl1⁺* lineage (**Figure 5D–E'''1–J''', O and P**). There were no differences in the mean diameter of the tongue or facial lymphatic vessels between the *Isl1^Cre/+*;*Prox1^fl/fl* homozygous mice and *Isl1^Cre/+*;*Prox1^fl/+* heterozygous mice (**Figure 5A–J, M and Q**). Next, we examined the contribution of the *Isl1⁺* lineage to regional lymphatic vessel formation by assessing the relative area of eGFP⁺ cells among VEGFR3⁺ LECs to determine whether the regional differences in the effects of *Isl1⁺*-lineage-specific *Prox1* deletion on lymphatic vessel formation were due to differences in compensation by other cell sources. *Isl1^Cre/+*;*Prox1^fl/+* heterozygous mice showed that the *Isl1⁺* lineage was almost totally responsible for the development of the tongue and facial skin lymphatic vessels (**Figure 5A–E''', N and R**). On the other hand, *Isl1^Cre/+*;*Prox1^fl/fl* homozygous mice showed a lower contribution of the *Isl1⁺* lineage to the facial skin lymphatics (**Figure 5F1–J''' and R**), whereas its contribution to the lymphatic vessels in the tongue was not decreased (**Figure 5F–H and N**). These results suggested that defects in LEC differentiation and/or maintenance due to *Prox1* deletion in the *Isl1⁺* lineage were compensated for by LECs from other cell sources, probably of venous origin, in facial skin, but not in the tongue.

## Loss of Prox1 in the *Tek*⁺ lineage confirms the heterogeneous origins of LECs

To further investigate the regional differences in the contributions of venous and non-venous cell sources to LECs, we crossed *Prox1^fl* mice with endothelial/hematopoietic cell-specific *Tek-Cre* mice. Immunostaining revealed decreased Prox1 expression in *Tek⁺* LECs in the back skin of *Tek-Cre;Prox1^fl/fl* homozygous mice compared to *Tek-Cre;Prox1^fl/+* heterozygous mice at E16.5, indicating efficient knockdown of Prox1, whereas *Tek⁻* LECs were observed similarly (**Figure 6—figure supplement 1A, B**). We also observed blood-filled lymphatic vessels in the back skin of the *Tek-Cre;Prox1^fl/fl* homozygous mice, indicating the formation of abnormal anastomosis between lymphatic and blood vessels due to Prox1 deficiency, as previously described (*Johnson et al., 2008*; **Figure 6—figure supplement 1A, B**). We then immunostained sagittal sections of the resultant embryos for PECAM, green fluorescent protein (GFP), and the LEC marker LYVE1 at E16.5, when lymphatic networks are distributed

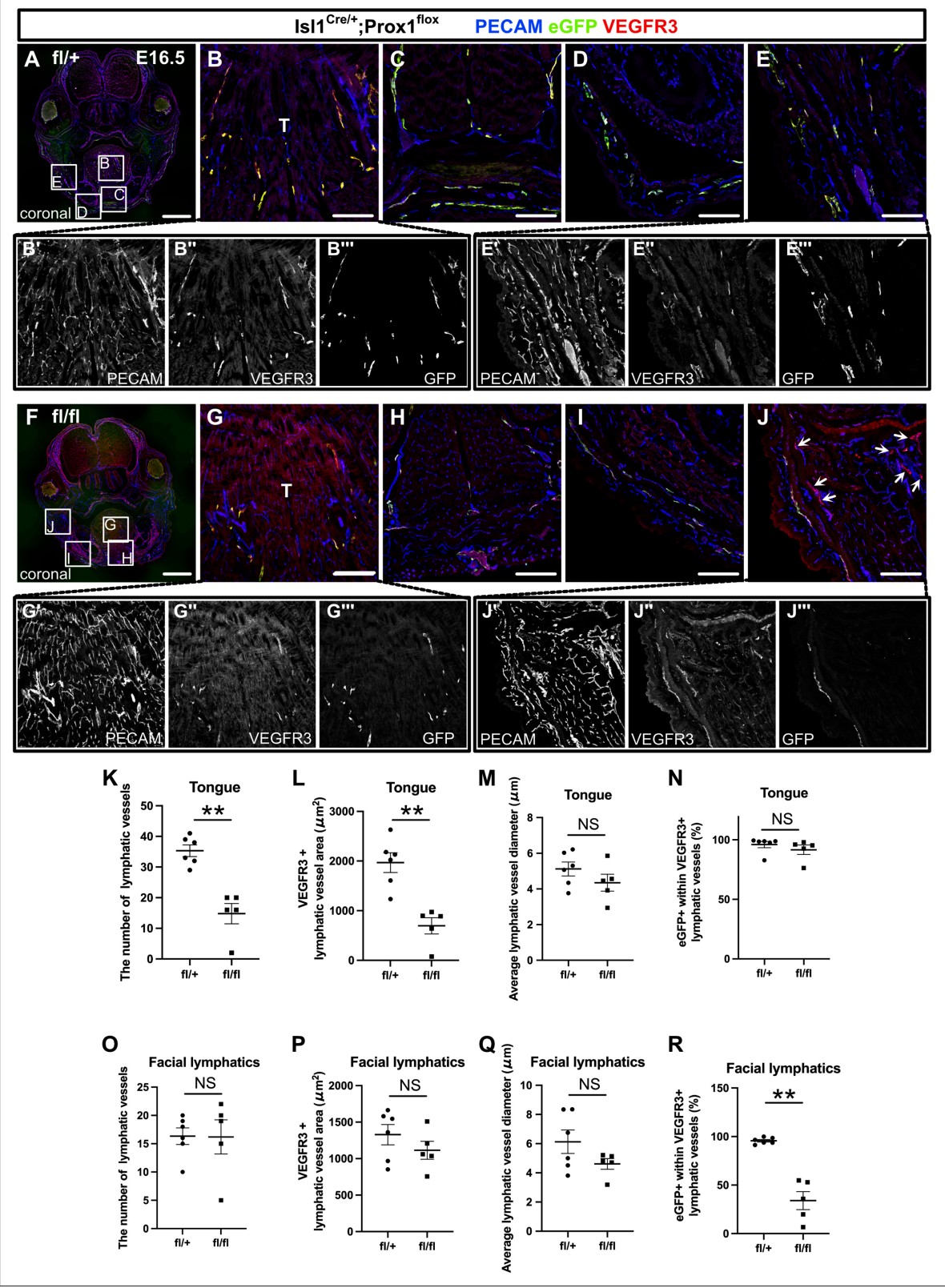

**Figure 5.** The inactivation of Prox1 in *Isl1*[+] lineages confirmed the contribution of the cardiopharyngeal mesoderm (CPM) to cranial lymphatic vessel development. (**A–J'''**) Coronal sections of *Isl1*[Cre/+];*Prox1*[fl/+] and *Isl1*[Cre/+];*Prox1*[fl/fl] mouse embryos, in which platelet endothelial cell adhesion molecule (PECAM), eGFP, and vascular endothelial growth factor receptor 3 (VEGFR3) were labeled at embryonic day (E) 16.5, are shown. (**J**) The number of eYFP[-]/ VEGFR3[+] lymphatic vessels in facial skin was increased in the *Isl1*[Cre/+]; *Prox1*[fl/fl] homozygous mice (white arrows). (**K–R**) The results of quantitative analysis

*Figure 5 continued on next page*

*Figure 5 continued*

of lymphatic vessel formation in the tongue (**K–N**) and facial skin (**O–R**) are shown. (**\*\*p=0.0043 (K, L, R**)). All the data are presented as the mean ± SEM, and statistical analyses were performed using the non-parametric Mann-Whitney U-test. Each dot represents a value obtained from one sample. \*\*p<0.01; T, tongue. Scale bars, 100 µm (**B–E, G–J**), 1 mm (**A, F**).

The online version of this article includes the following source data and figure supplement(s) for figure 5:

**Source data 1.** Quantification of lymphatic vessels phenotypes in *Isl1^{Cre/+}*;*Prox1^{fl/+}* and *Isl1^{Cre/+}*;*Prox1^{fl/fl}* mouse embryos at embryonic day (E) 16.5.

**Figure supplement 1.** Prox1 expression level was decreased in *Isl1^+* lymphatic endothelial cells (LECs) in *Isl1^{Cre/+}*;*Prox1^{fl/fl}* embryos.

**Figure supplement 1—source data 1.** Quantification of Prox1^+ cells in the first pharyngeal arch in *Isl1^{Cre/+}*; *Prox1^{fl/+}* and *Isl1^{Cre/+}*;*Prox1^{fl/fl}* mouse embryos at embryonic day (E) 12.5.

**Figure supplement 1—source data 2.** Quantification of the number of Prox1^+ cells in the tongue and the lower jaw of *Isl1^{Cre/+}*;*Prox1^{fl/+}* and *Isl1^{Cre/+}*;*Prox1^{fl/fl}* mouse embryos at embryonic day (E) 13.5.

throughout the whole body (*Srinivasan et al., 2007*). We identified LECs by their luminal structure and the colocalization of PECAM and LYVE1, the latter of which is also known to be expressed in a subset of macrophages.

We compared lymphatic vessel development in the tongue, the skin of the lower jaw, and back skin between mice that were heterozygous and homozygous for the *Prox1^{fl}* allele to assess the morphological changes in LYVE1^+/PECAM^+ lymphatic vessels seen in each tissue. In the tongue, the number of LYVE1^+/PECAM^+ lymphatic vessels was significantly lower and the mean lymphatic vessel diameter was significantly higher in the *Tek-Cre;Prox1^{fl/fl}* homozygous mice than in the *Tek-Cre;Prox1^{fl/+}* heterozygous mice (*Figure 6A and B* and *Figure 6—figure supplement 2A–C*). Almost all of the LYVE1^+/PECAM^+ lymphatic vessels in the tongue were positive for eGFP in the *Tek-Cre;Prox1^{fl/+}* heterozygous mice (*Figure 6A* and *Figure 6—figure supplement 2D*), indicating that the majority of LECs derived from *Isl1^+* CPM cells developed through *Tek* expression in the tongue. Remarkably, in the tongues of the *Tek-Cre;Prox1^{fl/fl}* homozygous mice, many of the eGFP^+ cells were not incorporated into the LYVE1^+/PECAM^+ lymphatics, resulting in an increased eGFP^+ area to lymphatics area ratio (*Figure 6A and B* and *Figure 6—figure supplement 2D*), which was indicative of impaired differentiation of venous cells into LECs or impaired maintenance of LEC identities due to a lack of Prox1 expression in the *Tek^+* endothelial cells.

On the contrary, compared with that seen in the *Tek-Cre;Prox1^{fl/+}* heterozygotes lymphatic vessel formation in the lower jaw and back skin was not significantly affected in the *Tek-Cre;Prox1^{fl/fl}* homozygotes, although the mean vessel diameter in the back skin of the homozygotes was increased (*Figure 6C–F* and *Figure 6—figure supplement 2E–G,I–K*). In contrast to the tongue, the eGFP^+ area to LYVE1^+/PECAM^+ lymphatics area ratios in the lower jaw and back skin were relatively low in both the *Tek-Cre;Prox1^{fl/+}* heterozygotes and *Tek-Cre;Prox1^{fl/fl}* homozygotes (*Figure 6C–F* and *Figure 6—figure supplement 2H, L*). These results demonstrate that the lymphatic vessels in these regions are composed of *Tek^+* and *Tek^−* LECs, supporting the idea that LECs have heterogeneous origins.

Taken together with the data from the experiments involving *Isl1^{Cre/+}* mice, these results suggest that the LECs in the craniofacial region have heterogeneous origins and developmental processes. Specifically, the LECs in the tongue are derived from the *Isl1^+/Tek^+* lineage, whereas the LECs in facial skin, including the skin on the lower jaw, are derived from the *Isl1^+/Tek^+* and *Isl1^+/Tek^−* lineages, which may compensate for each other when LECs from one lineage are impaired. Similarly, the LECs in back skin are derived from the *Tek^+* and *Tek^−* lineages, which may compensate for each other, whereas the *Isl1^+* lineage is not involved in the development of these cells.

## Discussion

In this study, we demonstrated that *Isl1^+* CPM cells, which are known to be progenitors of the cranial and cardiac musculature and connective tissue, contribute to the formation of lymphatic vessels in the cardiopharyngeal region, including the tongue, facial skin, larynx, and cardiac outflow tracts. Tamoxifen-inducible genetic lineage tracing further indicated that *Isl1^+* CPM-derived progenitors only have the potential to differentiate into LECs before E9.5. In accordance with these findings, *Isl1^+* CPM-derived LECs showed distinct spatiotemporal developmental processes and subsequently coordinated with *Isl1^−* venous-derived LECs arising from the lymph sac-forming domain to form the

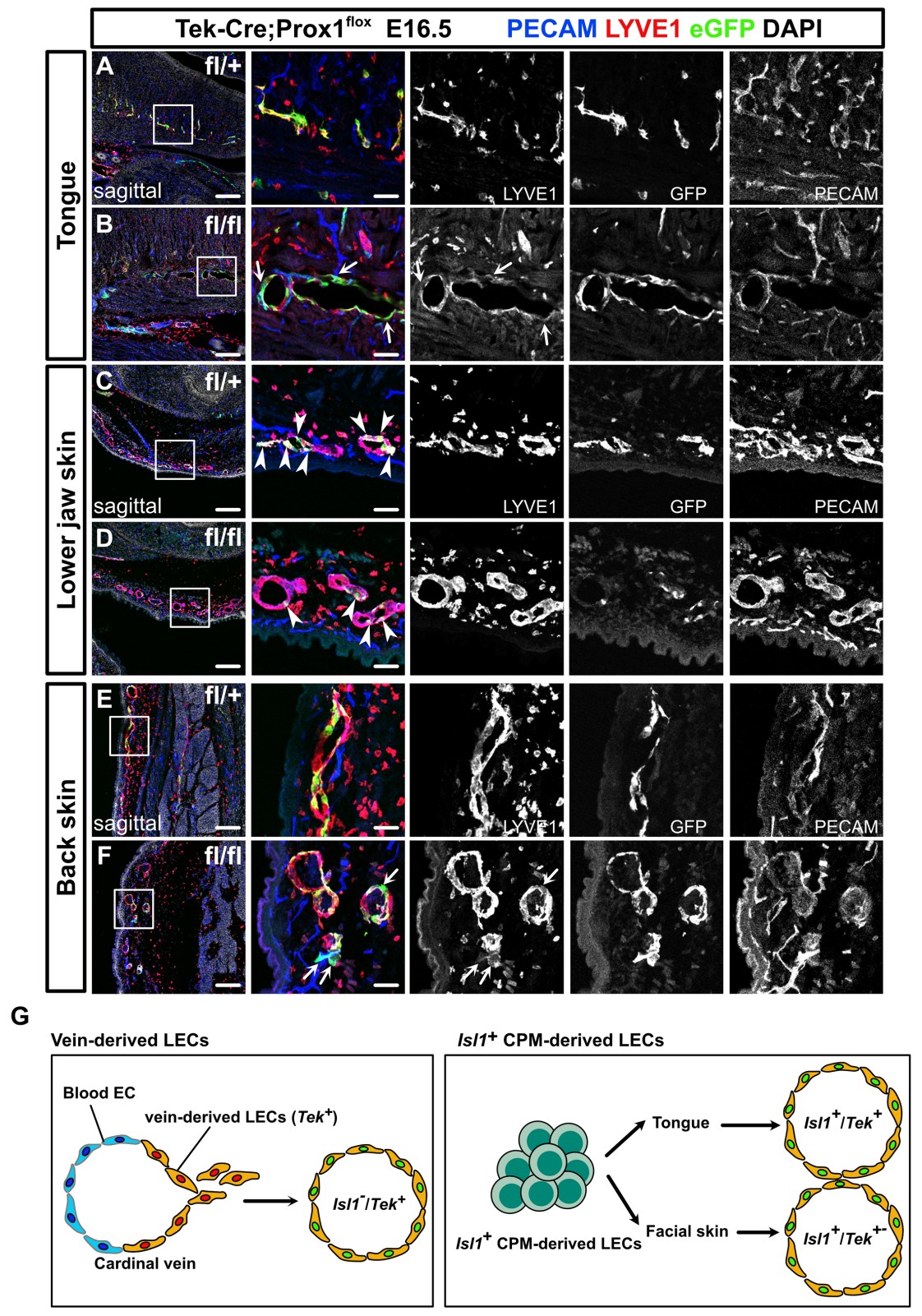

**Figure 6.** Prox1 knockdown in Tek+ lineages revealed regional differences in lymphatic vessel development. (**A–F**) Sagittal sections of *Tek-Cre;Prox1fl/+* or *Tek-Cre;Prox1fl/fl* mouse embryos, in which platelet endothelial cell adhesion molecule (PECAM), eGFP, lymphatic vessel endothelial hyaluronan receptor 1 (LYVE1), and DAPI were labeled at embryonic day (E) 16.5, are shown. (**A, B**) In the tongue, the number of LYVE1-/eGFP+/PECAM+ cells in lymphatic vessels was increased in the *Tek-Cre;Prox1fl/fl* embryos (white arrows). (**C, D**) In the skin of the lower jaw, the contribution of eGFP+ cells to lymphatic

*Figure 6 continued on next page*

*Figure 6 continued*

vessels was small in both the *Tek-Cre; Prox1^{fl/+}* and *Tek-Cre;Prox1^{fl/fl}* embryos (white arrows). (**E, F**) LYVE1⁻/eGFP⁺/PECAM⁺ cells were observed in the back skin of the *Tek-Cre;Prox1^{fl/fl}* embryos (white arrows). (**G**) Schematic representation of lineage classification in venous-derived LECs and *Isl1⁺* CPM-derived LECs. The LECs in the tongue are derived from the *Isl1⁺/Tek⁺* lineage, whereas the LECs in facial skin, including the skin on the lower jaw, are derived from the *Isl1⁺/Tek⁺* and *Isl1⁺/Tek⁻* lineages. Scale bars, 100 μm (**A–F**).

The online version of this article includes the following source data and figure supplement(s) for figure 6:

**Figure supplement 1.** Prox1 expression level was decreased in *Tek⁺* lymphatic endothelial cells (LECs) in *Tek-Cre;Prox1^{fl/fl}* embryos.

**Figure supplement 2.** Lymphatic vessel phenotypic differences between *Tek-Cre;Prox1^{fl/+}* and *Tek-Cre;Prox1^{fl/fl}* embryos.

**Figure supplement 2—source data 1.** Quantification of lymphatic vessels phenotypes in *Tek-Cre;Prox1^{fl/+}* and *Tek- Cre; Prox1^{fl/fl}* embryos at embryonic day (E) 16.5.

systemic lymphatic vasculature. In addition, conditional KO of *Prox1* in the *Isl1⁺* lineage resulted in lymphatic vessel deficiencies in the tongue. In contrast, in the facial skin, the loss of *Isl1⁺* LECs was compensated for by increased numbers of *Isl1⁻* LECs. Accordingly, conditional KO of *Prox1* in *Tek⁺* cells produced regional lymphatic phenotypic differences in the tongue, the skin of the lower jaw, and back skin. These results indicate that the LECs in the cardiopharyngeal region are mainly derived from *Isl1⁺* CPM progenitors (***Figure 7***).

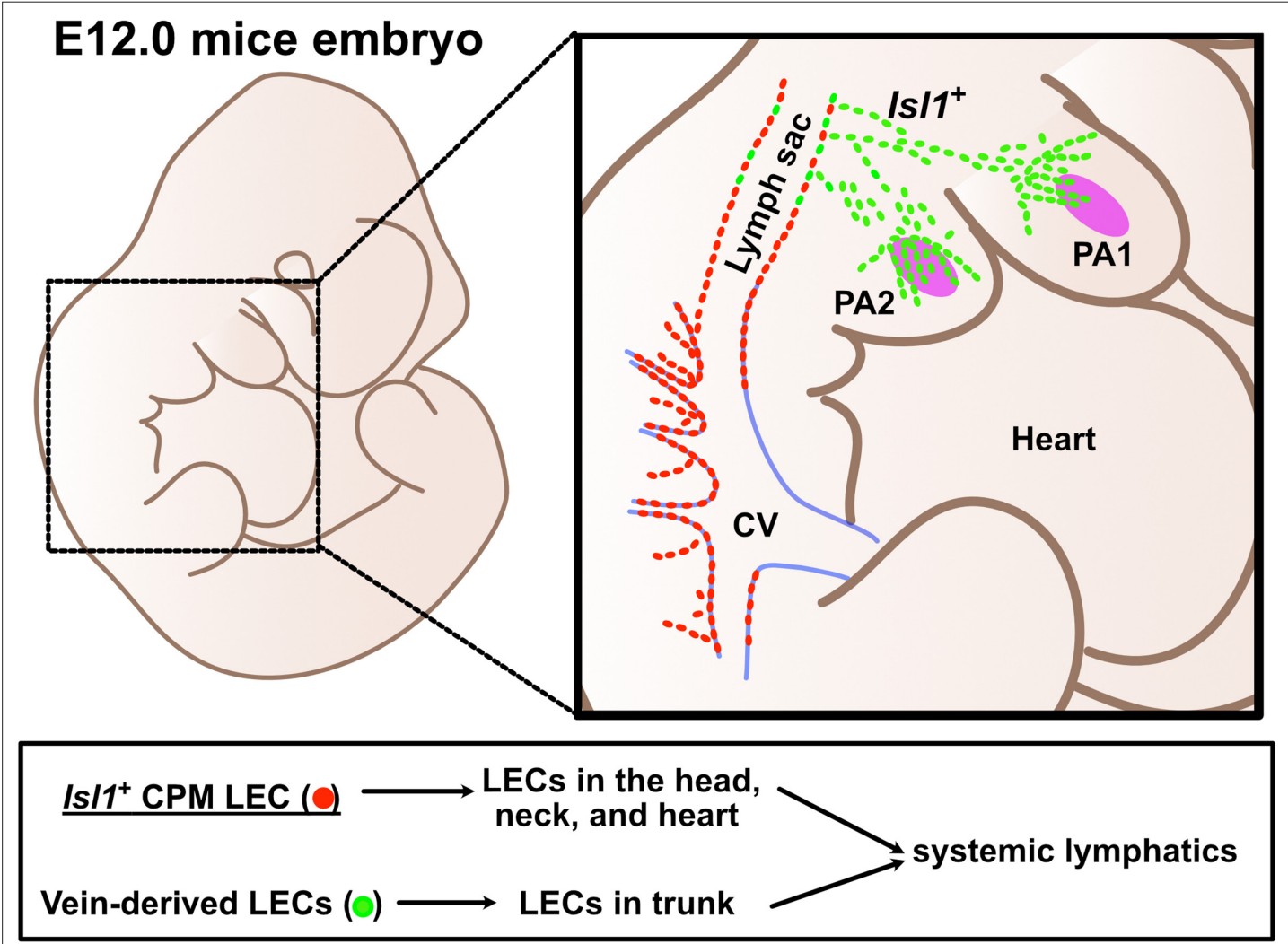

**Figure 7.** Lymphatic endothelial cells (LECs) are derived from two distinct origins. Schematic representation of the origins of LECs. LECs are mainly derived from cardinal veins (red dots) and the cardiopharyngeal mesoderm (CPM) (green dots).

The present study further indicates that the LECs in the tongue are derived from *Tek*-expressing cells among the *Isl1*⁺ lineage. Although it is unclear whether *Isl1*⁺-derived cells at the *Tek*-expressing stage represent a venous endothelial identity, this result means that *Tek*⁺ LECs are not equivalent to cardinal vein-derived LECs (***Figure 6G***). Furthermore, *Tek-Cre;Prox1^fl/fl^* homozygotes exhibited greater numbers of eGFP⁺/LYVE1⁻/PECAM⁺ cells than *Tek-Cre;Prox1^fl/+^* heterozygotes (***Figures 5A–C,F–H and 6A***). As a previous study found that conditional KO of *Prox1* could reprogram LECs to become blood endothelial cells (BECs) (***Johnson et al., 2008***), the number of BECs (eGFP⁺/LYVE1⁻/PECAM⁺ cells) may be increased in the tongues of *Tek-Cre;Prox1^fl/fl^* homozygous mice due to impaired maintenance of the LEC identity of *Isl1*⁺ CPM-derived cells that would have normally expressed Prox1. In contrast, the *Isl1*⁺ CPM-derived LECs in facial skin were not labeled by *Tek-Cre*, indicating that there is heterogeneity within *Isl1*⁺ CPM-derived LECs. Although further studies are needed to examine the phenotypic differences between the LECs in the tongue and facial skin, it is possible that differentiation processes may be affected by the extracellular environment, for example, through interactions with the surrounding cells and extracellular matrix.

A recent study has suggested that *Pax3*⁺ paraxial mesoderm-derived cells contribute to the cardinal vein and therefore venous-derived LECs originate from the *Pax3*⁺ lineage (***Stone and Stainier, 2019***). The same group has further argued that the *Pax3*⁺ lineage gives rise to lymphatic vessels on the trunk side through lymphvasculogenesis (***Lupu et al., 2022***). Therefore, the *Isl1*⁺ and *Pax3*⁺ lineages may complement each other to form systemic lymphatic vessels. In experiments involving *Myf5^Cre/+^* mice, it was also suggested that the *Myf5*⁺ lineage contributes to LECs in embryonic lymph sacs (***Stone and Stainier, 2019***); however, we could not identify *Myf5*⁺ LECs in *Myf5^CreERT2/+^* mice after tamoxifen treatment at E8.5 (***Figure 1—figure supplement 2***). This discrepancy may have been caused by differences in the mouse lines, the timing of Cre recombination, the genetic background, or the breeding environment.

Several studies have confirmed that *Isl1*⁺ LECs contribute to the ventral side of the heart (***Lioux et al., 2020***; ***Maruyama et al., 2019***). Milgrom-Hoffman et al. identified Flk1⁺/Isl1⁺ endothelial populations in the second pharyngeal arch in mouse embryos from E7.5 to E9.5 (***Milgrom-Hoffman et al., 2011***). These cell populations may serve as progenitors for LECs and BECs in the cardiopharyngeal region through the sequential expression of *Flk1*, *Tek*, and/or *Prox1* in response to different cues, leading to diverse fate determination, as revealed by recent single-cell RNA-sequence analyses (***Nomaru et al., 2021***; ***Wang et al., 2019***).

Heart development and pharyngeal muscle development are known to be tightly linked, suggesting that these tissues share common evolutionary origins (***Diogo et al., 2015***; ***Tzahor and Evans, 2011***). As the CPM is conserved in various species, CPM-derived LECs may also be evolutionarily conserved. In agreement with this, several studies involving zebrafish have indicated that facial and cardiac LECs originate from distinct cell sources from venous-derived LECs (***Eng et al., 2019***; ***Gancz et al., 2019***).

LMs are congenital lesions, in which enlarged and/or irregular lymphatic connections do not function properly. The causes of LMs are unknown, but LMs commonly occur in the head and neck regions (***Perkins et al., 2010***). Accumulating evidence has shown that mutations in the *phosphatidylinositol-4,5-bisphosphate 3-kinase catalytic subunit alpha* (*PIK3CA*) gene are found more frequently in LECs than in fibroblasts in LM patients (***Blesinger et al., 2018***). Given the fact that CPM-derived LECs were distributed around the head and neck and were found in the lymph sac-forming domain in the cervical region in the present study (***Figures 1–4***), LMs may be caused by *PIK3CA* mutations in CPM-derived LECs. In addition to LMs, several blood vessel malformations frequently occur in the neck and facial regions. Thus, some types of blood vessel malformations may also be caused by mutations in CPM progenitors.

In summary, the present study supports the idea that *Isl1*⁺ CPM cells are progenitors of LECs, which broadly contribute to cranial, neck, and cardiac lymphatic vessels in concert with venous-derived LECs (***Figure 7***). These findings are expected to shed new light on the cellular origins of lymphatic vessels and the molecular mechanisms of lymphatic vessel development, which may increase our understanding of the evolution of lymphatic vessels and the pathogenesis of lymphatic system-related diseases.

## Materials and methods
### Mouse strains
The following mouse strains were used: *Isl1^{Cre/+}* (*Cai et al., 2003*) (Cat# JAX:024242, RRID:IMSR_JAX:024242), *Isl1^{CreERT2/+}* (*Isl1^{MerCreMer/+}* is described as *Isl1^{CreERT2/+}*) (*Laugwitz et al., 2005*) (Cat# JAX:029566, RRID:IMSR_JAX:029566), *Wnt1-Cre* (*Jiang et al., 2000*) (Cat# JAX:007807, RRID:IMSR_JAX:007807), *Myf5^{CreERT2/+}* (*Biressi et al., 2013*) (Cat# JAX:023342, RRID:IMSR_JAX: 023342), *Rosa26^{eYFP/+}* (*Srinivas et al., 2001*) (Cat# JAX:006148, RRID:IMSR_JAX:006148), *Rosa26^{tdTomato/+}* (*Madisen et al., 2010*) (Cat# JAX:007914, RRID:IMSR_JAX:007914), *Tek-Cre* (*Kisanuki et al., 2001*) (Cat# JAX:008863, RRID: IMSR_JAX:008863), and *Prox1^{fl/+}*(*Iwano et al., 2012*). All mice were maintained on a mixed genetic background (C57BL/6J × Crl:CD1(ICR)), and both sexes were used (the mice were randomly selected). The genotypes of the mice were determined via the polymerase chain reaction using tail-tip or amnion DNA and the primers listed in *Supplementary file 1*. The mice were housed in an environmentally controlled room at 23 ± 2°C, with a relative humidity level of 50–60%, under a 12 hr light:12 hr dark cycle. Embryonic stages were determined by timed mating, with the day of the appearance of a vaginal plug being designated E0.5. All animal experiments were approved by the University of Tokyo (ethical approval number: H17-250) and Mie University (ethical approval number: 728) animal care and use committee, and were performed in accordance with institutional guidelines.

### Immunohistochemistry, histology, confocal imaging, and quantification
For the histological analyses, hearts and embryos were collected, fixed in 4% paraformaldehyde for 1 hr at 4°C, and stored in phosphate-buffered saline or embedded in optimal cutting temperature compound (Sakura Finetek, Tokyo, Japan). Sixteen-µm-thick frozen sections were stained with hematoxylin (Merck) and eosin (Kanto Chemical, Tokyo, Japan). Immunostaining of 16-µm-thick frozen sections was performed using primary antibodies against CD31 (553370, BD Pharmingen, RRID: AB_394816, 1:100), Flk1 (555307, BD Pharmingen, RRID:AB_395720, 1:100), Isl1 (AF1837, R&D systems, RRID:AB_2126324, 1:250), Prox1 (11-002, AngioBio, RRID: AB_10013720, 1:200; AF2727, R&D Systems, RRID: AB_2170716, 1:200), LYVE1 (11-034, AngioBio, 1:200; AF2125, R&D Systems, RPID: AB_2297188, 1:150), VEGFR3 (AF743, R&D Systems, RRID: AB_355563, 1:150), and GFP (GFP-RB-AF2020, FRL, RRID:AB_2491093, 1:500). Alexa Fluor-conjugated secondary antibodies (Abcam, RRID:AB_2636877, RRID:AB_2636997, RRID:AB_2752244, 1:400) were subsequently applied. The same protocol was followed for whole-mounted hearts and embryos, with the primary and secondary antibody incubation periods extended to two nights. Immunofluorescence imaging was conducted using a Nikon C2 confocal microscope or Keyence BZ-700. All images were processed using the ImageJ and Nikon NIS Elements software.

### Tamoxifen injection
For the lineage tracing using the *Isl1^{CreERT2/+}* or *Myf5^{CreERT2/+}* lines, tamoxifen (20 mg/mL; Sigma) was dissolved in corn oil. Pregnant mice were injected intraperitoneally with 125 mg/kg body weight of tamoxifen at the indicated timepoints.

### Statistical analysis
Data are presented as the mean ± standard error of the mean (SEM). Mann-Whitney U-tests were used for comparisons between two groups. p-Values of <0.05 were considered statistically significant. Data were analyzed using GraphPad Prism version 9 (GraphPad Software).

### Quantification of the section and whole-mount images
For the quantification of section immunostaining at E16.5 embryos, the average of two 16-µm-thick sections taken every 50 µm and ×10 power field of views (0.42 mm²/field) for each anatomical part (the larynx, the skin of the lower jaw, the tongue, and the cardiac outflow tracts) (*Figure 1H*, *Figure 2H and P*, and *Figure 2—figure supplement 1I*) or one middle section containing the same anatomical part (*Figure 6—figure supplement 2*) were subjected to the analyses. In the facial skin, lymphatic vessels in superficial layers of dermis were subjected to the analyses. The middle sagittal sections, including the aorta, larynx, and tongue, which were selected as hallmarks of midline, was chosen from

created sections. The coronal sections, including both eyes, tongue, and olfactory lobes with left and right symmetrical features, were selected and subjected to the analyses (*Figure 5K and R*). For E12.5 embryos (*Figure 4O*), two 16-µm-thick sagittal sections taken every 50 µm, including the first and second pharyngeal arches and outflow tracts, were subjected to analyses. The area and the number of cells were measured manually using ImageJ software. For the whole-mount immunostaining of embryos and the heart, the whole samples were scanned every 20 µm and confirmed eYFP contribution to LECs (*Figure 3*) and cardinal veins (*Figure 4J*, and *Figure 3—figure supplement 1B, D, F, H, J*).

## Acknowledgements

We thank all of the laboratory members for their helpful discussion and encouragement. This study was supported in part by Grants-in-Aid for Scientific Research from the Ministry of Education, Culture, Sports, Science, and Technology, Japan (19H01048 to HK and 20K17072 to KM); the Japan Foundation for Applied Enzymology (VBIC to KM); the Miyata Foundation Bounty for Pediatric Cardiovascular Research (KM); the SENSHIN Medical Research Foundation (KM); Takeda Science Foundation (KM); the Platform for Dynamic Approaches to Living Systems of the Ministry of Education, Culture, Sports, Science, and Technology, Japan; and the Core Research for Evolutional Science and Technology (CREST) program of the Japan Science and Technology Agency (JST), Japan (JPMJCR13W2 to HK).

## Additional information

### Funding

| Funder | Grant reference number | Author |
| --- | --- | --- |
| Core Research for Evolutional Science and Technology | JPMJCR13W2 | Hiroki Kurihara |

The funders had no role in study design, data collection and interpretation, or the decision to submit the work for publication.

### Author contributions

Kazuaki Maruyama, Conceptualization, Resources, Data curation, Formal analysis, Funding acquisition, Validation, Investigation, Visualization, Methodology, Writing - original draft, Project administration, Writing - review and editing; Sachiko Miyagawa-Tomita, Kyoko Imanaka-Yoshida, Supervision; Yuka Haneda, Investigation; Mayuko Kida, Data curation; Fumio Matsuzaki, Resources; Hiroki Kurihara, Supervision, Validation, Visualization, Writing - review and editing

### Author ORCIDs

Kazuaki Maruyama http://orcid.org/0000-0002-3935-328X
Sachiko Miyagawa-Tomita http://orcid.org/0000-0001-6646-8368
Fumio Matsuzaki http://orcid.org/0000-0001-7902-4520

### Ethics

All animal experiments were approved by the University of Tokyo (ethical approval number: H17-250) and Mie University (ethical approval number: 728) animal care and use committee, and were performed in accordance with institutional guidelines.

### Decision letter and Author response

Decision letter https://doi.org/10.7554/eLife.81515.sa1
Author response https://doi.org/10.7554/eLife.81515.sa2

## Additional files

### Supplementary files

• Supplementary file 1. Primers used for genotyping.

• MDAR checklist

## Data availability

All data generated or analysed during this study are included in the manuscript and supporting file; Source Data files have been provided for Figure 1—source data 1, Figure 2—figure supplement 1—source data 1, Figure 2—source data 1 and 2, Figure 4—source data 1, Figure 5—figure supplement 1—source data 1 and 2, Figure 5—source data 1, and Figure 6—figure supplement 2—source data 1.

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
