## [Editor Report]

This paper provides fundamental insight into the developmental source of lymphatic endothelial cells, which has been debated for over a century. This important work characterises the development of the mouse craniofacial lymphatics, and provides compelling evidence for a non-venous source of craniofacial lymphatic endothelial cells. The manuscript is well presented and will be of interest to developmental, vascular and lymphatic vascular biologists.

---

## [Decision Letter]

[Editors' note: this paper was reviewed by Review Commons.]

Thank you for submitting your article "The cardiopharyngeal mesoderm contributes to lymphatic vessel development" for consideration by *eLife*. Your article has been reviewed by 3 peer reviewers at Review Commons, and the evaluation at *eLife* has been overseen by a Reviewing Editor and Didier Stainier as the Senior Editor.

Based on the previous reviews and the revisions, the manuscript has been improved but there are some remaining issues that need to be addressed, as outlined below:

– The data presented in Figure 5 should be interpreted carefully as residual VEGFR3+ GFP+ cells likely indicate incomplete recombination of the Prox1 locus (probably just one allele) by the Isl1-Cre line, as a lack of PROX1 expression during differentiation should lead to an absence of LECs. Thus, it's likely that recombination of Prox1 is less efficient in the progenitors of LECs found in the lower jaw and cheeks than in the tongue. The expression of PROX1 protein should be assessed in these samples to understand the level of knockout achieved.

– Similarly, GFP expression in LYVE1 positive lymphatic vessels in Tie2-Cre;Prox1 fl/fl animals again suggests incomplete recombination of Prox1. The identity of the GFP+ cells that are not incorporated into lymphatic vessels should be clarified by imaging at higher magnification as currently, the data are not clear. The analyses in Figure 6 should be repeated with a more specific marker of lymphatic endothelial cells than LYVE1. Either PROX1 or VEGFR3. The PECAM1 staining in the sections presented in Figure 6A-B should be improved.

– It is likely that Myf5 is expressed transiently and at low levels in any mesodermal progenitor that gives rise to the endothelium. The Myf5-CreERT2 mouse line used in this study was constructed using an IRES-CreERT2 cassette in the 3'UTR of the Myf5 gene, meaning that expression of CreERT2 from this locus is likely significantly lower than in the Myf5-Cre line previously used to investigate LEC origins (Stone and Stainier, Dev Cell, 2019). Thus, the conclusions that can be drawn from the experiment presented in Supplementary figure 2 are limited. We recommend deleting the second sentence of the discussion 'We also showed that Myf5+ myogenic lineages, which were previously suggested to be possible sources of LECs35, did not contribute to lymphatic vasculature formation in Myf5-CreERT2 mice subjected to tamoxifen treatment at E8.5' and leaving the discussion of these analyses presented on lines 288-292, which more accurately place these data in the context of published work.

– Reviewer #3 at Review Commons suggested using a Pax3-Cre to assess the contribution of this lineage to facial lymphatics. These analyses have been published and so it would be sufficient to reference Stone and Stainier, Dev Cell, 2019.

– The following statement "The same group has further argued that the Pax3+ lineage gives rise to lymphatic vessels on the trunk side through lymphangiogenesis (Lupu et al., 2022)" should read "The same group has further argued that the Pax3+ lineage gives rise to lymphatic vessels on the trunk side through lymphvasculogenesis (Lupu et al., 2022)"

---

## [Author Response]

1. General StatementsReviewer #1 (Evidence, reproducibility and clarity (Required)):Please see combined review below in the next section,Reviewer #1 (Significance (Required)):This is a descriptive manuscript providing a few new insights into a well-recognized and biologically important phenomenon – the lymphatic endothelial cells have heterogeneous origins in different organs. Overall, the idea of Islet1 lineage contributes to regional lymphatic vessel formation during a particular developmental stage is exciting and proven with detailed and careful lineage tracing. The first observation that Islet1 lineage gives rise to cardiac lymphatic vessels was published by the same group in Dev Bio in 2019 so the novelty here is dampened, although the pharyngeal lymphatics and the exact time of these non-venous origin lymphatic vessels arise were not previously characterized – so the current manuscript does provide new important insights. Both the data quality and manuscript layout need improvements, especially when it comes to defining where Islet1 is expressed at all the stages and statistics.The following suggestions will deepen the scope of the manuscript:

First of all, we would like to express our appreciation to the reviewer for all the constructive comments. We carefully read the reviewer’s comments and discussed it. We agree with the reviewer that our manuscript needs improvements with changes in layout several additional experiments. We have also included several description and new immunostaining data (e.g., Isl1,VEGFR3 and LYVE1 co-staining), to confirm our new findings and highlight the importance of the current manuscript beyond our previous one in Dev Biol in 2019. We also have included detailed quantification methods, single-channel images with improved data resolution, and improved clarity of the manuscript.

2. Description of the planned revisionsPoint 4.The effect of lineage-specific Prox1 knockout is very descriptive, without any discussion of the potential biological function of such cellular origin heterogeneity. This part may be worth a few follow-up experiments in later embryonic stages or even in postnatal stages. The authors demonstrated that loss of Prox1 in Islet1 lineage decreases the number of lymphatic vessels and leads to lymphangiectasia, but whether this phenotype can be later compensated or shows any clinical impact was not proven. Therefore, the statement made in line 206 is questionable, and whether Islet1 lineage-derived lymphatic endothelial cells are dispensable/indispensable remains unclear.

We agree with the reviewer in that additional follow up experiments using later embryonic or postnatal stages will give an insight into the potential biological function of cellular origin heterogeneity. We are generating lineage-specific Prox1 knockout mice by treating *Isl1MerCreMer; Prox1^fl/fl^* mice with tamoxifen at E8.5 to analyze phenotypes in the facial lymphatic vessels.

3. Description of the revisions that have already been incorporated in the transferred manuscriptPoint 1 and 21.Whether Isl1 lineage is independent of venous-derived endothelial cells.2.This point is very important: the manuscript does not actually show Isl1 expression through the stages they are inducing. I would want to be sure that lymphatic endothelial cells at this stage don't express Isl1. Another way to get at this is to maybe use other second-heart fields or even broader mesoderm drivers that are known to be never expressed in endothelial cells to confirm the findings.

In our previous work (Maruyama et al., *Dev Biol* 452:134–143, 2019, Figure 2C), we demonstrated that *Isl1*^+^ lineages using *Isl1-Cre* mice did not contribute to endothelial cells in the cardinal vein and its branches (intersomitic vessels: ISVs), which had been thought to the primary and biggest sources of lymphatic endothelial cells (LECs). In this paper, we confirmed this finding using *Isl1MerCreMer* mice with tamoxifen treatment at E8.5 (Figure 4J). We have scanned whole embryos and detected no eYFP^+^ cells in the cardinal vein or ISVs (the detailed quantification methods have been added in Methods section). Consistently, another group (Lioux et al., *Dev Cell* 52:350-363, 2020) re-evaluated this point using *Isl1-Cre* mice that the *Isl1*^+^ lineage contribute to endothelial cells of the cardinal vein only by less than 2%, which neither explains the abundant contribution of the *Isl1*^+^ lineage to coronary lymphatics (>50%) nor its restriction to the ventral heart. Based on these reports, we supposed that the *Isl1*^+^ lineage was independent of LECs derived from the cardinal vein and ISVs.

In the revised manuscript, we added new data showing thorough expression patterns of Isl1, Prox1, Flk1, and PECAM in the E9.0 to E11.5 pharyngeal arches and cardinal veins by immunostaining and presented them as Supplemental Figure 4. In these sections, we detected Isl1 and Prox1 expression with partial overlapping within the pharyngeal mesodermal core, whereas Isl1 was co-expressed with Flk1, or PECAM neither in vessel-like structures around the mesodermal core nor in the cardinal vein and their surrounding Prox1^+/^PECAM^+^ LECs (Supplemental Figure 4H’ and J’) confirming the independency. These findings have been described in the manuscript as follows:

“To identify possible Isl1^+^ LEC progenitors, we investigated the expression patterns of Isl1, Prox1, and vascular endothelial markers (Flk1 and PECAM) by immunostaining sections of E9.0 to E11.5 pharyngeal arches and cardinal veins. Consistent with the previous report (Cai et al., 2003), Isl1 was abundantly expressed in the core mesoderm of the first and second pharyngeal arches corresponding to the CPM from E9.0 to E11.5 (Nathan et al., 2008), where Prox1^+^ cells also aggregated and partially overlapped with Isl1 signals (Supplemental Figure 4A, A’ C, C’ E, E’ G, G’ I, I’). By contrast, Flk1^+^ or PECAM^+^ cells were distributed mainly around the CPM and not expressed Isl1 (Supplemental Figure 4A, A’ C, C’ E, E’ G, G’ I, I’). Furthermore, Isl1 was expressed neither in the endothelial layer of the cardinal vein nor in surrounding Prox1^+^/PECAM^+^ LECs (Supplemental Figure 4B, B’ D, D’, F, F’, H, H’, J, and J’). Taken together with the result from Myf5-CreERT2 mice, these results indicate that Isl1^+^ non-myogenic CPM cells may serve as LEC progenitors independent of venous-derived LECs and the commitment to LEC differentiation occurs before E9.5 in the pharyngeal arch region.” (Page 6-7, lines 187-200)

Point 3.The author stated that Islet1 lineage gives rise to lymphatic endothelial cells via the Tie2 mechanism but did not elaborate on this part. What is the potential relationship between Islet1 and Tie2? Or Tie2 just serves as a pan-endothelial lineage marker here?

To clearly demonstrate the relationship between *Isl1*^+^ and *Tie2*^+^ lineages in facial lymphatics, we added schematic representation in Figure 6G, which showed the differential Tie2 expression in lymphatic vessels in the tongue and facial skin.

Related to point 1 and 2, it has been thought that almost all LECs are formed from cardinal vein-derived *Tie2*^+^ endothelial cells. However, we identified the presence of *Isl1*^+^/*Tie2*^+^ LECs in the tongue, which are apparently not originated from the cardinal vein. In previous reports using Tie2-GFP mice or in situ hybridization of *Tie2*, *Tie2* was not detected in the developing LECs at E9.5, 11.5, 13.5, and E15.5 (Motoike et al., 2000; Srinivasan et al., 2007). In adult mice, *Tie2* expression in lymphatics was only observed in restricted regions (Morisada et al., 2005; Tammela et al., 2005). Taken together with our present data that the differentiation fate of *Isl1*^+^ CPM-derived LECs was determined between E6.5 and E9.5 (Figure 2-4, Supplemental Figure 3), Tie2 is supposed to be transiently expressed during LEC differentiation in the tongue from early *Isl1*^+^ CPM cells, although it remains difficult to identify the Tie2-expressing stage during non-venous LEC differentiation.

It will be an important future subject to identify the stage and implication of transient Tie2 expression in the lineage and, in this paper, we want to just note that the *Tie2*^+^ lineage does not always mean the derivation from cardinal vein endothelial cells.

This point has already been included in the manuscript as follows:

“The present study further indicates that the LECs in the tongue are derived from Tie2-expressing cells among the Isl1^+^ lineage. Although it is unclear whether Isl1^+^-derived cells at the Tie2expressing stage represent a venous endothelial identity, this result means that Tie2^+^ LECs are not equivalent to cardinal vein-derived LECs.” (Page 10, 298-301)

Minor Point 1.The layout of the manuscript needs to be reorganized:Details in statistical methods and quantification logic were completely missing from the manuscript. For example, definitions of "a sample" (how many sections are taken from one biological sample and how many fields take from one section, etc.), "number of vessels per field", "diameters", and of what parameters the numbers were normalized to, etc. need to be described in the Materials and methods section. For instance, it is not clear how "tomato+ lymphatic vessels per field/Vegfr3+ lymphatic vessels" was defined. First, what proportion of tomato+ cells need to colocalize with Vegfr3 expression cells in a specific vessel to make this vessel being determined as a "tomato+ lymphatic vessel"? Most data provided here are section immunostaining where "multiple vessels" are very likely coming from different cross-sections of one same vessel in the same field. Second, Vegfr3 can stain venous endothelial cells in earlier stages so the specificity of this marker can be controversial. These are some important technical aspects to include in the revised version. Figures needing more description in quantification methods include but are not limited to Figure 1H, 2H, 2P, 5K-R.

We have revised the statistical methods from the ratio of the count of the number to ratio of the area of lymphatic vessels in Figure 1H, 2H, P, and Supplemental Figure 3I to represent more precisely the contribution of Tomato^+^ cells to lymphatic vessels. We also added more detailed description of the quantification methods in ‘Materials and methods’ section, as follows:

“Quantification of the section and whole mount images

For the quantification of section immunostaining at E16.5 embryos, the average of two 16-μmthick sections taken every 50 μm and 10x power field of views (0.42 mm^2^/field) for each anatomical part (the larynx, the skin of the lower jaw, the tongue, and the cardiac outflow tracts) were subjected to the analyses. In the facial skin, lymphatic vessels in superficial layers of dermis were subjected to the analyses. The middle sagittal sections, including the aorta, larynx, and tongue, which were selected as hall marks of midline, was chosen from created sections. The coronal sections, including both eyes, tongue, and olfactory lobes with left and right symmetrical features, was selected. For E12.5 embryos (Figure 4O), two 16-μm-thick sagittal sections taken every 50 μm, including the 1^st^ and 2^nd^ pharyngeal arches and outflow tracts, were subjected to analyses. The area and the number of cells were measured manually using ImageJ software. For the whole mount immunostaining of embryos and the heart, the whole samples were scanned every 20 μm and confirmed eYFP contribution to LECs (Figure 3) and cardinal veins (Figure 4J, and Supplemental Figure 4B, D, F, H, J).” (Page 13, lines 402-416)

We also have tested expression patterns of VEGFR3 with Prox1 or LYVE1 as Supplemental Figure 1. At E14.5, VEGFR3 was widely co-expressed with Prox1 in the tongue, facial skin, and around the pulmonary artery (Supplemental Figure 1A-C’). At E16.5, VEGFR3 was co-expressed with LYVE1 in the tongue, facial skin, and around the pulmonary artery. Thus, we thought that VEGFR3 could be used as a marker of LECs in these cardiopharyngeal region.

This point has been included in the manuscript as follows:

“Co-immunostaining of platelet endothelial cell adhesion molecule (PECAM) and vascular endothelial growth factor receptor 3 (VEGFR3), which we confirmed its co-localization with lymphatic vessel endothelial hyaluronan receptor 1 (LYVE1) at E14.5 and E16.5 (Supplemental Figure 1), revealed tdTomato^+^ LECs in and around the larynx, the skin of the lower jaw, the tongue, and the cardiac outflow tracts, at various frequencies, whereas no such cells were found on the dorsal side of the ventricles, which agrees with our previous study (Maruyama et al., 2019).” (Page 4, lines 112-119)

Minor point 2.Data resolution needs to be improved. The magnification of the figures in Figure 1-4 is not sufficient to demonstrate the marker colocalization as described in the texts. Single-channel images (such as the ones shown in Figure 5-6 but in higher magnifications) are also necessary to show the coexpression of markers.

There was a limit on the data capacity when submitting the manuscript. We were therefore obliged to reduce the quality of images and file size. We have revised the figures to add several higher magnification and single-channel images with improved data resolution throughout Figure 1-4.

Minor point 3.The experimental design is not well-elaborated in the context. For example, the scientific logic of choosing a particular time point/stage for lineage-knockout induction or sample collection needs to be justified. Also, it seems that the authors are using fl/+ as control littermates in most of the experiments. Any specific reason favors using fl/+ heterozygous instead of fl/fl littermates without cre exposure, which is the more commonly used control sample in this kind of comparison, should be addressed.

Knockdown of Prox1 in the *Tie2*^+^ lineage has shown to cause an initial failure in specification of LECs at E14.5 with no appearance of lymphatics even at E17.5(Klotz et al., 2015; Lioux et al., 2020; Maruyama et al., 2019), indicating that the effect on lymphatic vessels would not be compensated even at E16.5. In addition, the systemic lymphatic network formation is almost completed at E16.5(Srinivasan et al., 2007), and the lineage trace was also evaluated at this stage. Thus, it was reasonable to compare the phenotype at E16.5.

This point has been addressed in the text as follows:

“When Prox1 is knocked down in the Tie2^+^ lineage, an initial failure in specification of LECs was confirmed at E14.5 with a lack of LECs even at E17.5(Klotz et al., 2015; Lioux et al., 2020; Maruyama et al., 2019). Therefore, we compared lymphatic vessel phenotypes at E16.5, by which systemic lymphatics formation is normally completed (Srinivasan et al., 2007).” (Page 7, lines 208212)

In *Prox1-flox*(Prox1^fl/+^) mice, recombinant cells were labeled with EGFP(Iwano et al., 2012), as already described in the manuscript (Page 7, lines206-208). Therefore, the recombined cells can be visualized by EGFP expression in both heterozygous (fl/+) and homozygous (fl/fl) mice, which enables phenotype analysis referring to the recombined (knocked-out in fl/fl) cells. Importantly, these mice showed no specific phenotypes (Klotz et al., 2015; Maruyama et al., 2019). It is therefore reasonable to use heterozygous mice as controls to compare the phenotype appropriately. Although fl/fl littermates without cre exposure could usually serve as controls, they do not express EGFP in the Prox1 lineage, detracting from their utility (Klotz et al., 2015; Maruyama et al., 2019).

Minor point 4.Some of the phrases are not clear in the text- either because of the writing style or because the corresponding figures failed to support the statements. These include but are not limited to lines 104-106, 122, 206, 226, and 228-233.104-106: we crossed Isl1-Cre mice, which express Cre recombinase under the control of the Isl1 promoter and in which second heart field-derivatives are effectively labeled, with the transgenic reporter line R26R-tdTomato at E16.5.

We have re-phrased this sentence as follows:

“We crossed Isl1-Cre mice, which express Cre recombinase under the control of the Isl1 promoter and in which second heart field-derivatives are effectively labeled, with the transgenic reporter line R26R-tdTomato and analyzed at E16.5, when lymphatic networks are distributed throughout the whole body.” (Pages 4, lines 109-112)

122: After tamoxifen was administered at E8.5, tdTomato^+^ cells were broadly detected in the muscle in the head and neck regions at E16.5, indicating effective Cre-mediated recombination of the target gene.

We have re-phrased this sentence as follows:

“After tamoxifen was administered at E8.5, tdTomato^+^ cells were broadly detected in the skeletal muscle in the head and neck regions at E16.5, indicating effective Cre recombination in CPMderived musculatures.” (Page 4, lines 130-132)

We have also included red arrowheads, indicating CPM-derived musculatures in Supplemental Figure 2.

206: These results suggested that defects in LEC differentiation and/or maintenance due to Prox1 deletion in the Isl1^+^ lineage were compensated for by other cell sources, probably of venous origin, in facial skin, but not in the tongue, resulting in impaired lymphatic vessel formation in the tongue.

We have re-phrased this sentence as follows:

“These results suggested that defects in LEC differentiation and/or maintenance due to Prox1 deletion in the Isl1^+^ lineage were compensated for by LECs from other cell sources, probably of venous origin, in facial skin, but not in the tongue.” (Page 7-8, lines 231-233)

226: Almost all of the LYVE1^+^/PECAM^+^ lymphatic vessels in the tongue were positive for eGFP in the Tie2-Cre;Prox1^fl/+^ heterozygous mice (Figure 6A and Supplemental Figure 3D), indicating that the majority of LECs derived from Isl1^+^ CPM cells developed through Tie2 expression in the tongue.

We have added new cartoon in Figure 6G to more clearly show the relation of Tie2 expression in *Isl1*^+^ lineages. Previous reports have used Tie2-Cre mice to show the vein-derived LECs (Klotz et al., 2015; Srinivasan et al., 2007), because most of cardinal vein endothelium were composed of *Tie2*^+^ lineages. In our present study, in the tongue, most of the LECs were derived from *Isl1*^+^/Tie2^+^ lineages (Figure 1D, H, Figure 2D, Figure 4G, Figure 5B, N, Figure 6A and Supplemental Figure 3F, I). These data suggested that there was a group of *Tie2*^+^ lineages even though they are derived from non-venous *Isl1*^+^ lineages.

Minor point 5.Reference needed for Myf5-Cre as a driver for Myogenic CPM in the Results section.

We have included several reference, as shown below:

(Harel et al., 2012, 2009; Heude et al., 2018)

Harel et al., *Dev cell*, 2009

Minor point 6.In discussion the reference to Pitx2-driven mesenteric lymphatic heterogeneity (Mahadevan et al. 2014) is missing yet Islet1 has been shown downstream of Pitx2 (Davis et al. 2008). The authors should discuss their findings of gut lymphatic heterogeneity in this context, considering that mediastinum is mesentery-derived.

*Isl1*^+^ CPM-derived LECs have been distributed to the anterior mediastinum and their relationship to mesenteric lymphatic vessels, which continuous with the thoracic duct in the posterior mediastinum, is currently unclear. However, since this paper is valuable for understanding the heterogeneity of the origins of LECs, we have included the indicated paper (Mahadevan et al., 2014) in ‘Introduction’ section to show gut lymphatic heterogeneity. (Page 3, line 67)

4. Description of analyses that authors prefer not to carry outRelated to point 4.Regarding the use of other second-heart field drivers as the reviewer recommended,

Lioux et al., have already shown the contribution of *Mef2c-AHF*^+^, which marked CPM-derived cells including second heart field, cranial musculatures and connective tissues (Adachi et al., 2020), to ventral cardiac lymphatics. We are also trying to introduce *Mef2c-AHF-Cre* mice, but it is unfortunately delayed due to the pandemic of COVID-19.

Reviewer #2 (Evidence, reproducibility and clarity (Required)):The manuscript entitled “The cardiopharyngeal mesoderm contributes to lymphatic vessel development” identified a novel non-venous origin of craniofacial and cardiac LECs using genetic lineage tracing. Their results also revealed the spatiotemporal difference between CPM- and venous-derived LECs. Overall, the paper is well-organized and has certain implications for understanding lymphatic development. However, some issues still need to be improved:Reviewer #2 (Significance (Required)):This study enriched the contribution of CPMs to broader regions of the facial, cardiac and laryngeal lymphatic network and revealed the spatiotemporally difference between CPM- and venous-derived LECs, which provided some basic reference for understanding lymphatic vessel development.

First of all, we would like to express our appreciation to the reviewer for all the constructive comments. We carefully read the reviewer’s comments and discussed it. We agree with the reviewer’s comments to make the text easier to understand and emphasize what we really want to say.

2. Description of the revisions that have already been incorporated in the transferred manuscriptPoint1. Clearly, the introduction needs to be more concise and focused on the main questions you propose to answer and why these questions are important.

We have revised introduction section to be more concise and focus on the developmental process of lymphatic vessels and its relation to CPM. (Page 2-4, lines 41-103)

2. In the Discussion section, you should focus on how the questions have been answered and what they mean. And it would be rash to infer the role of LECs in lymphatic malformation. It would be helpful to validate the changes of CPM-derived LECs in LM patient samples.

We have revised the Discussion section to be more concise. To demonstrate our findings more clearly, we have also revised and added some cartoons in Figure 6G and Figure 7.

3. For the statistical analysis, all the quantitative data should be tested for statistical significance. There are several bar charts lacking P values.

We have included P values in the Figure legends.

Reviewer #3 (Evidence, reproducibility and clarity (Required)):Short summary of the findings and key conclusions:The work from Murayama and colleagues traces the ontogenetic origin of the endothelial cells of the lymphatic vessels in the head and neck region. Using the Cre-lox-based mouse genetics approach, they conclude that the lymphatic endothelial cells (LECs) in this region have mixed origin, with contributions from both the cardiopharyngeal mesoderm (CPM) as well as from cardinal vein. The lineage tracing study is buttressed by assaying LEC formation following selective deletion of the key LEC regulator Prox1 in CPM lineage.Reviewer #3 (Significance (Required)):The nature and significance of the advance for the field & the work in the context of the existing literature:Groups working in the domain of cardiopharyngeal mesoderm (CPM) have focussed on skeletal muscle and heart development. This pool is also known to give rise to skeletal tissues as well as blood vessel endothelium. A recent work Nomaru et al. (Morrow group, Nat Commun 2021) has identified a multi-lineage primed population in the cardiopharyngeal field. In this context, the work from Maruyama and colleagues highlights the versatility of CPM by providing evidence for the emergence of LEC from this multipotent pool. This complex developmental potential of CPM has implications to understand the evolutionary origin of CPM itself.The connective tissues in the head/neck have mixed origins (Heude et al., 2018 and Grimaldi et al. 2022 from Tajbakhsh group)- from CPM as well as neural crest. This work shows mixed origin for LECs. These works begin to put together the pieces of the puzzle of vertebrate head evolution. Jacob proposed evolution is tinkering. This appears to be true both at the molecular level as well as the cellular level. Head tissues appear to have been put together by exploiting varied sources.The study is of broad interest to developmental biologists.Reviewer: A developmental biologist with an interest in understanding the axial patterning of mesoderm early during mammalian development. Not an expert in lymphatic vasculature development.

First of all, we would like to express our appreciation to the reviewer for all the constructive comments. We carefully read the reviewer’s comments and discussed it.

2. Description of the revisions that have already been incorporated in the transferred manuscriptIn addition, the article should be revised to include the number of sections and the number of cells counted per embryo in the Figure legend in each case. This will help assess how robust and reliable are the measurements.

We have revised the statistical methods from the ratio of the count of the number to the area in Figure 1H, 2H, P, and Supplemental Figure 3I to demonstrate more precisely the contribution of Tomato^+^ cells in lymphatic vessels. We also added more detailed description of the quantification methods in ‘Materials and methods’ section, as follows:

Quantification of the section and whole mount images

“For the quantification of section immunostaining at E16.5 embryos, the average of two 16-μmthick sections taken every 50 μm and 10x power field of views (0.42 mm^2^) for each anatomical part (the larynx, the skin of the lower jaw, the tongue, and the cardiac outflow tracts) were subjected to the analyses. In the facial skin, lymphatic vessels in superficial layers of dermis were subjected to the analyses. The middle sagittal sections, including the aorta, larynx, and tongue, which were hall marks of midline, was selected from created sections. The coronal sections, including both eyes, tongue, and olfactory lobes with left and right symmetrical features, was selected. For E12.5 embryos (Figure 4O), two 16-μm-thick sagittal sections taken every 50 μm, including the 1^st^ and 2^nd^ pharyngeal arches and outflow tracts, were subjected to analyses. The area and the number of Prox1^+^ cells were measured manually using ImageJ software. For the whole mount immunostaining of embryos and the heart, the whole samples were scanned every 20 μm and confirmed eYFP contribution to LECs (Figure 3) and cardinal veins (Figure 4J, and Supplemental Figure 4B, D, F, H, J).” (Pages 13, lines 402-416)

We have also included the number of eYFP^+^/Prox1^+^ cells among Prox1^+^ cells in the first and second pharyngeal in the Figure 4O legends as follows;

“(the number of eYFP^+^/Prox1^+^ cells (10.83 (mean) ± 1.249 (SEM)): Prox1^+^ cells (30.83 ± 4.549)) or E9.5 (the number of eYFP^+^/Prox1^+^ cells (2.833 ± 1.108): Prox1^+^ cells (35.50 ± 5.847)).” (Page 23, lines 684-686)Minor comments 1.Several groups have contributed to the CPM literature. The citation of seminal works from Tzahor and Kelly groups is good, however, work from other groups has not been cited. For example, reports such as Heude et al. and Grimaldi et al. from Tajbakhsh group are very relevant to this work.

According to reviewer’s suggestion, we have included following references in the introduction section for the explanation of CPM derivatives. (P3, line 77)

(Heude et al., 2018)

Minor comments 2.It would help the reader if the authors explain the reasons for selecting specific regions, such as the tongue, and the skin of the lower jaw, for the study.

This is because many lymphatic vessels are distributed in these cardiopharyngeal area and these area is well known as anatomical parts where lymphatic malformation most often occurs. This has been mentioned in the manuscript as follows:

*From a clinical viewpoint, head and neck regions contributed by the CPM are the most common sites of lymphatic malformations (LMs)* (Page 3-4, lines 99-100)

3. Description of analyses that authors prefer not to carry outPoint 1.The key conclusions: LECs in the head and neck region derive from CPM. LECs in this region have mixed developmental origins. Both these conclusions are convincingly supported by the study. However, the work would greatly be strengthened by Pax3-Cre lineage tracing. This would complement the Isl1-Cre lineage tracing. As the authors observe, the LEC descendants of Isl1+ cells also appear to go through Tie2+ state. Therefore, Tie2-Cre study has not helped to delineate the LECs of CPM and cardinal vein origins. In this context, tracing with Pax3-Cre is likely to give a very clear picture of LEC origins.

We agree with the reviewer in that the data using Pax3-Cre mice will strengthen our manuscripts. Unfortunately, we could not find out researchers who had this line in our society in Japan. For using this line, we need to get cryo-recovered mice from Jaxon laboratory. It will take at least several months. Therefore it is not realistic for us to use Pax3-Cre mice in this work because of time limitation. Instead, we addressed this issue by rewriting the discussion on the possible complementation with the Pax3-Cre lineage by citing (Lupu et al., 2022; Stone and Stainier, 2019).

This point has been addressed in the text as follows:

“A recent study has suggested that Pax3^+^ paraxial mesoderm-derived cells contribute to the cardinal vein and therefore venous-derived LECs originate from the Pax3^+^ lineage (Stone and Stainier, 2019). The same group has further argued that the Pax3^+^ lineage gives rise to lymphatic vessels on the trunk side through lymphangiogenesis(Lupu et al., 2022). Therefore, the Isl1^+^ and Pax3^+^ lineages may complement each other to form systemic lymphatic vessels.” (Page 10, lines 314-319)

Minor comments 3.The authors should consider presenting the wholemount images, such as those in Figures 3A and 3E for Figures 5 and 6. This would help assess the lymphatic vessel development in a holistic manner.

Although we tried to do the whole mount images of facial and tongue lymphatics, we could not succeed. Antibodies did not penetrate well on the tongue and, as for lymphatics of facial skin, their complicated morphology prevented clear visualization. Whole-mount imaging of the entire head was difficult for the same reason. In our experience, the antibody was useful for immunostaining of the early-stage embryos (up to E11.5) and the surface area of the heart, where lymphatic vessels were distributed on the epicardium. Even in the whole-mount heart, we have not succeeded in clear and estimable imaging of the vascular structure in the myocardium. Instead, we improved the quality of images and statistical comparisons in the revised manuscript, which we believe makes it more convincing.

References.

Adachi N, Bilio M, Baldini A, Kelly RG. 2020. Cardiopharyngeal mesoderm origins of musculoskeletal and connective tissues in the mammalian pharynx. *Development*

147:dev185256. doi:10.1242/dev.185256

Cai C-L, Liang X, Shi Y, Chu P-H, Pfaff SL, Chen J, Evans S. 2003. Isl1 Identifies a Cardiac

Progenitor Population that Proliferates Prior to Differentiation and Contributes a Majority of

Cells to the Heart. *Dev Cell* 5:877–889. doi:10.1016/s1534-5807(03)00363-0

Grimaldi A, Comai G, Mella S, Tajbakhsh S. 2022. Identification of bipotent progenitors that give rise to myogenic and connective tissues in mouse. *ELife* 11:e70235. doi:10.7554/*eLife*.70235

Harel I, Maezawa Y, Avraham R, Rinon A, Ma H-Y, Cross JW, Leviatan N, Hegesh J, Roy A, Jacob-Hirsch J, Rechavi G, Carvajal J, Tole S, Kioussi C, Quaggin S, Tzahor E. 2012.

Pharyngeal mesoderm regulatory network controls cardiac and head muscle morphogenesis. *Proc National Acad Sci* 109:18839–18844. doi:10.1073/pnas.1208690109

Harel I, Nathan E, Tirosh-Finkel L, Zigdon H, Guimarães-Camboa N, Evans SM, Tzahor E. 2009. Distinct Origins and Genetic Programs of Head Muscle Satellite Cells. *Dev Cell*

16:822–832. doi:10.1016/j.devcel.2009.05.007

Heude E, Tesarova M, Sefton EM, Jullian E, Adachi N, Grimaldi A, Zikmund T, Kaiser J, Kardon G, Kelly RG, Tajbakhsh S. 2018. Unique morphogenetic signatures define mammalian neck muscles and associated connective tissues. *ELife* 7:e40179. doi:10.7554/*eLife*.40179

Klotz L, Norman S, Vieira JM, Masters M, Rohling M, Dubé KN, Bollini S, Matsuzaki F, Carr CA, Riley PR. 2015. Cardiac lymphatics are heterogeneous in origin and respond to injury. *Nature* 522:62–67. doi:10.1038/nature14483

Lioux G, Liu X, Temiño S, Oxendine M, Ayala E, Ortega S, Kelly RG, Oliver G, Torres M.

2020. A Second Heart Field-Derived Vasculogenic Niche Contributes to Cardiac Lymphatics. *Dev Cell* 52:350–363. doi:10.1016/j.devcel.2019.12.006

Lupu I-E, Kirschnick N, Weischer S, Martinez-Corral I, Forrow A, Lahmann I, Riley PR, Zobel T, Makinen T, Kiefer F, Stone OA. 2022. Direct specification of lymphatic endothelium from non-venous angioblasts. *Biorxiv* 2022.05.11.491403. doi:10.1101/2022.05.11.491403 Mahadevan A, Welsh IC, Sivakumar A, Gludish DW, Shilvock AR, Noden DM, Huss D, Lansford R, Kurpios NA. 2014. The Left-Right Pitx2 Pathway Drives Organ-Specific Arterial and Lymphatic Development in the Intestine. *Dev Cell* 31:690–706. doi:10.1016/j.devcel.2014.11.002

Maruyama K, Miyagawa-Tomita S, Mizukami K, Matsuzaki F, Kurihara H. 2019. Isl1expressing non-venous cell lineage contributes to cardiac lymphatic vessel development. *Dev Biol* 452:134–143. doi:10.1016/j.ydbio.2019.05.002

Morisada T, Oike Y, Yamada Y, Urano T, Akao M, Kubota Y, Maekawa H, Kimura Y, Ohmura M, Miyamoto T, Nozawa S, Koh GY, Alitalo K, Suda T. 2005. Angiopoietin-1 promotes

LYVE-1-positive lymphatic vessel formation. *Blood* 105:4649–4656. doi:10.1182/blood2004-08-3382

Motoike T, Loughna S, Perens E, Roman BL, Liao W, Chau TC, Richardson CD, Kawate T, Kuno J, Weinstein BM, Stainier DYR, Sato TN. 2000. Universal GFP reporter for the study of vascular development. *Genesis* 28:75–81. doi:10.1002/1526-968x(200010)28:2<75::aidgene50>3.0.co;2-s

Nathan E, Monovich A, Tirosh-Finkel L, Harrelson Z, Rousso T, Rinon A, Harel I, Evans SM, Tzahor E. 2008. The contribution of Islet1-expressing splanchnic mesoderm cells to distinct branchiomeric muscles reveals significant heterogeneity in head muscle development. *Development* 135:647–57. doi:10.1242/dev.007989

Srinivasan RS, Dillard ME, Lagutin OV, Lin F-J, Tsai S, Tsai M-J, Samokhvalov IM, Oliver G. 2007. Lineage tracing demonstrates the venous origin of the mammalian lymphatic vasculature. *Gene Dev* 21:2422–2432. doi:10.1101/gad.1588407

Stone OA, Stainier DYR. 2019. Paraxial Mesoderm Is the Major Source of Lymphatic

Endothelium. *Dev Cell* 50:247-255.e3. doi:10.1016/j.devcel.2019.04.034

Tammela T, Saaristo A, Lohela M, Morisada T, Tornberg J, Norrmén C, Oike Y, Pajusola K, Thurston G, Suda T, Yla-Herttuala S, Alitalo K. 2005. Angiopoietin-1 promotes lymphatic sprouting and hyperplasia. *Blood* 105:4642–4648. doi:10.1182/blood-2004-08-3327

[Editors' note: further revisions were suggested prior to acceptance, as described below.]

Based on the previous reviews and the revisions, the manuscript has been improved but there are some remaining issues that need to be addressed, as outlined below:– The data presented in Figure 5 should be interpreted carefully as residual VEGFR3+ GFP+ cells likely indicate incomplete recombination of the Prox1 locus (probably just one allele) by the Isl1-Cre line, as a lack of PROX1 expression during differentiation should lead to an absence of LECs. Thus, it's likely that recombination of Prox1 is less efficient in the progenitors of LECs found in the lower jaw and cheeks than in the tongue. The expression of PROX1 protein should be assessed in these samples to understand the level of knockout achieved.

To address recombination efficiency of the Prox1 locus and Prox1 expression levels in *Isl1*^+^ LECs, we have performed whole-mount and section immunostaining with PECAM, eGFP, and Prox1. In whole-mount immunostaining at E12.5, eGFP^+^/Prox1^+^ cells were observed in the PA1 of *Isl1^Cre/+^;Prox1^fl/+^* heterozygous mice, whereas the number of Prox1^+^ cells were decreased in *Isl1^Cre/+^;Prox1^fl/fl^* homozygous mice (Figure 5 —figure supplement 1A-F). (The numbers of Prox1^+^ cells in the first pharyngeal arch were 71.5 (mean) ± 3.5 (SEM) and 7.0 ± 1.0 in *Isl1^Cre/+^;Prox1^fl/+^* (n=2) and *Isl1^Cre/+^;Prox1^fl/fl^* (n=2) embryos, respectively). Thus, the recombination of the Prox1 locus occurred efficiently. At E13.5, eGFP^+^/Prox1^+^ cells were almost disappeared in the tongue, whereas eGFP^+^/Prox1^+^ cells were still observed in the lower jaw in *Isl1^Cre/+^;Prox1^fl/fl^* homozygous mice (Figure 5 —figure supplement 1G-P). Notably, the number of homozygous mice was reduced in comparison to that of heterozygous mice at E13.5 (11 heterozygous vs 3 homozygous), indicating that the homozygous genotype was partially lethal and embryos with incomplete recombination might survive.

This point is described as following:

“To test the recombination efficiency of the Prox1 locus in Isl1^+^ LECs, we performed whole-mounted and section immunostaining with PECAM, eGFP, and Prox1 in Isl1^Cre/+^;Prox1^fl/+^ heterozygous and Isl1^Cre/+^;Prox1^fl/fl^ homozygous mice at E12.5 and E13.5. At E12.5, eGFP^+^/Prox1^+^ cells were observed in the PA1 of Isl1^Cre/+^;Prox1^fl/+^ heterozygous mice, whereas the number of Prox1^+^ cells was decreased and most of eGFP^+^ cells were negative for Prox1 in Isl1^Cre/+^;Prox1^fl/fl^ homozygous mice (Figure 5 —figure supplement 1A-F), indicating efficient knockdown of Prox1. At E13.5, eGFP^+^/Prox1^+^ cells was almost diminished in the tongue, whereas eGFP^+^/Prox1^+^ cells were still observed in the lower jaw in Isl1^Cre/+^;Prox1^fl/fl^ homozygous mice (Figure 5 —figure supplement 1G-P). This discrepancy may indicate that the recombination efficiency differs among tissues and that embryos with low recombination efficiency could survive until E13.5. (Page 7, lines211-222)”.

– Similarly, GFP expression in LYVE1 positive lymphatic vessels in Tie2-Cre;Prox1 fl/fl animals again suggests incomplete recombination of Prox1. The identity of the GFP+ cells that are not incorporated into lymphatic vessels should be clarified by imaging at higher magnification as currently, the data are not clear. The analyses in Figure 6 should be repeated with a more specific marker of lymphatic endothelial cells than LYVE1. Either PROX1 or VEGFR3. The PECAM1 staining in the sections presented in Figure 6A-B should be improved.

Regarding the point above, we have performed section immunostaining of *Tek-Cre;Prox1^fl/fl^* homozygous (n=3) and *Tek-Cre;Prox1^fl/+^* heterozygous (n=3) embryos for PECAM, eGFP, and Prox1 at E16.5 in the back skin. The Prox1 expression in *TeK^+^* LECs was diminished in *Tek-Cre;Prox1^fl/fl^* homozygous embryos compared to *Tek-Cre;Prox1^fl/+^* heterozygous embryos, whereas *Tek*^–^ LECs were observed (Figure 6 —figure supplement 1A-B). Interestingly, blood-filled lymphatic vessels were frequently observed in *Tek-Cre;Prox1^fl/fl^* homozygous embryos. Considering a previous report that loss of Prox1 resulted in the formation of anastomosis between blood and lymphatic vessels (Johnson et al., 2008), our observation may indicate the formation of anastomosis between blood and lymphatic vessels. They also indicated that loss of Prox1 in LECs altered marker expression patterns (VE-cadherin and Lyve1 expression were down-regulated in *CAGGCre-ERT2;Prox1^fl/fl^* mice (Figure 5C and D)). Thus, our observations that loss of LYVE1 expression in the *TeK^+^* LECs in *Tek-Cre;Prox1^fl/fl^* homozygous embryos were consistent with their report (Johnson et al., 2008).

This point was described as follows:

“Immunostaining revealed decreased Prox1 expression in TeK^+^ LECs in the back skin of Tek-Cre;Prox1^fl/fl^ homozygous mice compared to Tek-Cre;Prox1^fl/+^ heterozygous mice at E16.5, indicating efficient knockdown of Prox1, whereas Tek^–^ LECs were observed similarly (Figure 6 —figure supplement 1A and B). We also observed blood-filled lymphatic vessels in the back skin of the Tek-Cre;Prox1^fl/fl^ homozygous mice, indicating the formation of abnormal anastomosis between lymphatic and blood vessels due to Prox1 deficiency, as previously described (Johnson et al., 2008) (Figure 6 —figure supplement 1A and B).” (Page 8, lines248-255)

We also have improved PECAM immunostaining of Figure 6A and B. We also magnified Figure 6 to show more clearly the correlation of Prox1 knockdown and the decreased LYVE1 expression.

– It is likely that Myf5 is expressed transiently and at low levels in any mesodermal progenitor that gives rise to the endothelium. The Myf5-CreERT2 mouse line used in this study was constructed using an IRES-CreERT2 cassette in the 3'UTR of the Myf5 gene, meaning that expression of CreERT2 from this locus is likely significantly lower than in the Myf5-Cre line previously used to investigate LEC origins (Stone and Stainier, Dev Cell, 2019). Thus, the conclusions that can be drawn from the experiment presented in Supplementary figure 2 are limited. We recommend deleting the second sentence of the discussion 'We also showed that Myf5+ myogenic lineages, which were previously suggested to be possible sources of LECs35, did not contribute to lymphatic vasculature formation in Myf5-CreERT2 mice subjected to tamoxifen treatment at E8.5' and leaving the discussion of these analyses presented on lines 288-292, which more accurately place these data in the context of published work.

We have deleted indicated sentence.

– Reviewer #3 at Review Commons suggested using a Pax3-Cre to assess the contribution of this lineage to facial lymphatics. These analyses have been published and so it would be sufficient to reference Stone and Stainier, Dev Cell, 2019.

We have already included this article as follows:

“A recent study has suggested that Pax3^+^ paraxial mesoderm-derived cells contribute to the cardinal vein and therefore venous-derived LECs originate from the Pax3^+^ lineage (Stone and Stainier, 2019).” (Pages 10-11, lines 328-330)

– The following statement "The same group has further argued that the Pax3+ lineage gives rise to lymphatic vessels on the trunk side through lymphangiogenesis (Lupu et al., 2022)" should read "The same group has further argued that the Pax3+ lineage gives rise to lymphatic vessels on the trunk side through lymphvasculogenesis (Lupu et al., 2022)"

We have corrected lymphangiogenesis as lymphvasculogenesis.

References:

Johnson NC, Dillard ME, Baluk P, McDonald DM, Harvey NL, Frase SL, Oliver G. 2008. Lymphatic endothelial cell identity is reversible and its maintenance requires Prox1 activity. *Gene Dev* 22:3282–3291. doi:10.1101/gad.1727208

Maruyama K, Miyagawa-Tomita S, Mizukami K, Matsuzaki F, Kurihara H. 2019. Isl1-expressing non-venous cell lineage contributes to cardiac lymphatic vessel development. *Dev Biol* 452:134–143. doi:10.1016/j.ydbio.2019.05.002